# A hierarchical 3D-motion learning framework for animal spontaneous behavior mapping

Kang Huang [1,2,4], Yaning Han[1,2,4], Ke Chen[1,2], Hongli Pan[1], Gaoyang Zhao [1,2], Wenling Yi[1,2], Xiaoxi Li[1,2], Siyuan Liu[3], Pengfei Wei [1,2✉] & Liping Wang [1,2✉]

Animal behavior usually has a hierarchical structure and dynamics. Therefore, to understand how the neural system coordinates with behaviors, neuroscientists need a quantitative description of the hierarchical dynamics of different behaviors. However, the recent end-to-end machine-learning-based methods for behavior analysis mostly focus on recognizing behavioral identities on a static timescale or based on limited observations. These approaches usually lose rich dynamic information on cross-scale behaviors. Here, inspired by the natural structure of animal behaviors, we address this challenge by proposing a parallel and multi-layered framework to learn the hierarchical dynamics and generate an objective metric to map the behavior into the feature space. In addition, we characterize the animal 3D kinematics with our low-cost and efficient multi-view 3D animal motion-capture system. Finally, we demonstrate that this framework can monitor spontaneous behavior and automatically identify the behavioral phenotypes of the transgenic animal disease model. The extensive experiment results suggest that our framework has a wide range of applications, including animal disease model phenotyping and the relationships modeling between the neural circuits and behavior.

[1] Shenzhen Key Lab of Neuropsychiatric Modulation and Collaborative Innovation Center for Brain Science, Guangdong Provincial Key Laboratory of Brain Connectome and Behavior, CAS Center for Excellence in Brain Science and Intelligence Technology, Brain Cognition and Brain Disease Institute (BCBDI), Shenzhen-Hong Kong Institute of Brain Science-Shenzhen Fundamental Research Institutions, Shenzhen Institute of Advanced Technology, Chinese Academy of Sciences, Shenzhen, China. [2] University of Chinese Academy of Sciences, Beijing, China. [3] Pennsylvania State University, University Park, PA, USA. [4]These authors contributed equally: Kang Huang, Yaning Han. ✉email: pf.wei@siat.ac.cn; lp.wang@siat.ac.cn

The structure of animal behavior follows a bottom-up hierarchy constructed by time-varying posture dynamics, which has been demonstrated to be classical in ethological theory[1,2] and recent animal studies[3–6]. Such behavioral organization is considered to coordinate with neural activities[7,8]. Previous studies[9–11] using large-scale neuronal recordings have provided preliminary evidence from the neural implementation perspective. As the central goal of modern neuroscience, fully decoding this cross-scale dynamic relationship requires comprehensive quantification of neural activity and behavior. Over the past few decades, scientists have been working on improving the accuracy and throughput of neural dynamics manipulation and capturing. Meanwhile, for behavior quantification, there has been a revolution from simple behavioral parameters extraction to machine-learning (ML)-based behavior sequence recognition[12,13]. However, most previous methods[14,15] often emphasized feature engineering and pattern recognition for mapping raw data to behavioral identities. These black-box approaches lack the interpretability of cross-scale behavioral dynamics. Thus, it is a challenging task, but with a strong demand, to develop a general-purpose framework for the dynamic decomposition of animal spontaneous behavior.

Previous researchers addressed this challenge mainly from two aspects. The first aspect is behavioral feature capturing. Conventional animal behavior experiments usually use a single-camera top-view recording to capture the motion signal of behaving animals, leading to occlusions of the key body parts (e.g., paws), and these are very sensitive to viewpoint differences[16]. Thus, it is very challenging for the single-camera technology to capture the three-dimensional (3D) motion and then map the spontaneous behavior in a dynamic way. The recent emergence of ML toolboxes[17–19] has dramatically facilitated the animal pose estimation with multiple body parts. Thus, it enables us to study the animal kinematics more comprehensively and provides potential applications for capturing 3D animal movements. The second aspect is decomposing continuous time-series data into understandable behavioral modules. Previous studies on lower animals such as flies[10,20–22], zebrafishes[4,23–25], and *Caenorhabditis elegans*[26–28] utilized ML strategies and multivariate analysis to detect action sequences. However, mammalian behavior is highly complicated. Besides locomotion, animals demonstrate non-locomotor movement (NM) with their limbs (e.g., grooming, rearing, turning), and their organs have high dimensional[29–31] and variable spatiotemporal characteristics. Even for similar behaviors, the duration and composition of postural sequences vary. To define the start and end boundaries to segment continuous data into behavioral sequences, many ML-based open-source toolboxes[21] and commercial software do excellent work in feature engineering. They usually compute per-frame features that refer to position, velocity, or appearance-based features. The sliding windows technology then converts them into window features to reflect the temporal context[14,15]. Although these approaches effectively identify specific behaviors, behavior recognition becomes problematic when the dynamics of particular behaviors cannot be represented by the window features.

The present study proposes a hierarchical 3D-motion learning framework to address our contribution to these challenges. First, we acquired the 3D markerless animal skeleton with tens of body parts by the developed flexible and low-cost system. Through the systematic validations, we proved that our system could solve the critical challenges of body occlusion and view disappearance in animal behavior experiments. Second, aiming at the parallel and hierarchical dynamic properties of spontaneous behavior, we proposed a decomposition strategy preserving the behavior's natural structure. With this strategy, the high-dimensional, time-varying, and continuous behavioral series can be represented as various quantifiable movement parameters and low-dimensional behavior map. Third, we obtained a large sample of the *Shank3B*[−/−] mouse disease model data resources with our efficient framework. The results showed that our framework could detect behavioral biomarkers that have been identified previously and discover potential new behavioral biomarkers. Finally, together with the further group analysis of the behavioral monitoring under different experimental apparatus, lighting conditions, ages, and sexes, we demonstrated our framework could contribute to the hierarchical behavior analysis, including postural kinematics characterization, movement phenotyping, and group level behavioral patterns profiling.

## Results

**Framework of hierarchical 3D-motion learning.** In our framework, first we collect the animal postural feature data (Fig. 1a). These data can be continuous body parts trajectories that comprehensively capture the motion of the animal's limbs and torso, and they inform the natural characteristics of locomotion and NM. Locomotion can be represented by velocity-based parameters. NM is manifested by movement of the limbs or organs without movement of the torso and is controlled by dozens of degrees of freedom[32]. Hence, we adopted a parallel motion decomposition strategy to extract features from these time-series data independently (Fig. 1b, c). A two-stage dynamic temporal decomposition algorithm was applied to the centralized animal skeleton postural data to obtain the NM space. Finally, together with the additional velocity-based locomotion dimension, unsupervised clustering was used to reveal the structure of the rodent's behavior.

Our framework has two main advantages. First, it addresses the multi-timescale of animal behavior[33]. Animal behavior is self-organized into a multi-scale hierarchical structure from the bottom up, including poses, movements, and ethograms[34,35]. The poses and movements are low- and intermediate-level elements[36], while higher-level ethograms are stereotyped patterns composed of movements that adhere to inherent transfer rules in certain semantic environments[37]. Our two-stage pose and movement decomposition focuses on extracting the NM features of the first two layers. Second, our framework emphasizes the dynamic and temporal variability of behavior. The most critical aspect of unsupervised approaches is to define an appropriate metric for quantifying the relationship between samples. However, the duration and speed of NM segments of the same cluster may differ. To address this, we used a model-free approach called dynamic time alignment kernel (DTAK) as a metric to measure the similarity between the NM segments and thus equip the model to automatically search repeatable NM sequences. We then apply the uniform manifold approximation and projection (UMAP)[38] algorithm to visualize high-dimensional NM representations. After combining the locomotion dimension with NM space (Fig. 1c), we adopted hierarchical clustering to re-cluster the components and map the behavior's spatial structure (Fig. 1d).

**Collecting mouse motion data with a 3D multi-view motion-capture system.** To efficiently and comprehensively characterize the kinematics of free-moving animals, we developed a 3D multi-view motion-capture system (Fig. 2a, b) based on recent advances in techniques for pose estimation[17] and 3D skeletal reconstruction[39]. The most critical issues in 3D animal motion capture are efficient camera calibration, body occlusion, and viewpoint disappearance, which have not been optimized or verified[12]. To address these issues, we developed a multi-view video capture device (Supplementary Fig. 2a). This device

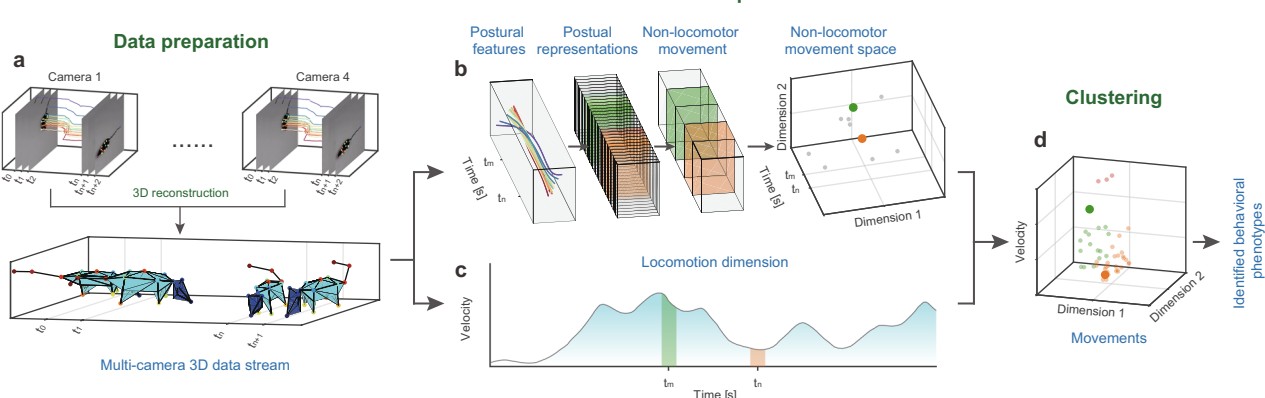

**Fig. 1 Hierarchical 3D-motion learning framework for animal behavior analysis. a** Data preparation: (1) image streams captured from four cameras with different 2D views; (2) animal body parts are tracked to generates separate 2D skeletal trajectories (color-coded traces); (3) reconstructing 3D body skeleton by integrating these four data streams. **b** Two-stage NM decomposition to generate the feature space: (1) pose decomposition groups continuous skeleton postural data into discrete postural sequences); (2) NM decomposition, two highlighted (green and orange) blocks represent two NMs decomposed from the postural sequences; (3) NM sequences mapped to their 2D features space (right), where each dot on the 3D axis corresponds to the NM block on the left. **c** Calculation of locomotion dimension. The continuous velocity of the behaving animal is first calculated, then average the velocity of each segment obtained in the NM decomposition step. **d** 3D scatter plot represents the combined NM and locomotion feature space. All the movements are clustered into three types (red, green, and orange dots) with the unsupervised approach.

integrates the behavioral apparatus, an auto-calibration module (Supplementary Fig. 2b, d), and synchronous acquisition of multi-view video streams (Supplementary Fig. 2c). While the conventional manual method requires half an hour to produce the required checkerboard for calibration, the auto-calibration module can be completed in 1 min.

We collected the naturalistic behavioral data of free-moving mice in a featureless circular open field (Supplementary Fig. 2a and Supplementary Movie M1). We analyzed the mouse skeleton as 16 parts (Fig. 2c) to capture the movements of the rodent's head, torso, paws, and tail. The following motion quantification did not involve the motion features of two parts of the tail. The data obtained from tracking representative mouse poses tracking (Fig. 2c) include the 3D coordinates ($x$, $y$, and $z$) of the body parts, which reveal that the high-dimensional trajectory series exhibits periodic patterns within a specific timescale. We next investigated whether the 3D motion-capture system could reliably track the animal in cases of body-part occlusion and viewpoint disappearance. We checked the DeepLabCut (DLC) tracking likelihood in the collated videos (0.9807 ± 0.1224, Supplementary Fig. 4a) and evaluated the error between the estimated two-dimensional (2D) body parts of every training set frame and the ground truth (0.534 ± 0.005%, Supplementary Fig. 5b). These results indicated that in most cases, four cameras were available for 2D pose tracking. Since 3D reconstruction can be achieved as long as any two cameras obtain the 2D coordinates of the same point in 3D space from different views, the reconstruction failure rate caused by body occlusion and viewpoint disappearances is determined by the number of available cameras. Therefore, we evaluated the average proportion of available cameras in situations of body part occlusion and viewpoint disappearance. The validation results for body-part occlusion show an average reconstruction failure rate of only 0.042% due to body occlusion or inaccurate body-part estimation (Supplementary Fig. 5c). While for viewpoint disappearances, both tests (Supplementary Fig. 6 and Supplementary Movies M4 and M5) proved that our system has a high reconstruction rate for animal body parts. Moreover, the artifact detection and correction features can recover the body parts that failed to be reconstructed. We calculated an overall reconstruction quality (0.9981 ± 0.0010, Fig. 2d) to ensure that the data were qualified for downstream analysis.

**Decomposing non-locomotor movements with dynamic time alignment kernel.** Conceptually, behavior adheres to a bottom-up hierarchical architecture (Fig. 3a)[34,35], and research has focused on elucidating behavioral component sequences contained in stimuli-related ethograms[40]. The purpose of the two-stage NM decomposition is to bridge the low-level vision features (postural time-series) to high-level behavioral features (ethograms). The first stage of the decomposition involves extracting postural representations from postural feature data. Since the definition of NM does not involve the animal's location or orientation, we pre-processed these data through center alignment and rotation transformation (Supplementary Fig. 7). Animal movement is continuous, and due to the high dimensionality of the mammalian skeleton, the behaviorally relevant posture variables are potentially infinite in number[12]. However, adjacent poses are usually highly correlated and redundant for behavior quantification and analysis[1], which is particularly evident in long-term recording. Therefore, for computational efficiency, we adopted a temporal reduction algorithm to merge adjacent, similar poses as postural representations in a local time range.

In the second stage, NM modules are detected from temporal reduced postural representations. Unlike the static property of poses, mammalian movements have high dimensionality and large temporal variability[41]: e.g., the contents, phases, and durations of the three pose sequences were not the same (Fig. 3a). Hence, we adopted a model-free approach to dynamically perform temporal aligning and cluster the temporally reduced postural representation data (Fig. 3b)[42]. This problem is equivalent to providing a $d$-dimensional time-series $X \in \Re^{d \times n}$ of animal postural representations with $n$ frames. Our task decomposes $X$ into $m$ NM segments, each of which belongs to one of the corresponding $k$ behavioral clusters. This method detects the change point by minimizing the error across segments; therefore, dynamic temporal segmentation becomes a problem of energy minimization. An appropriate distance metric is critical for modeling the temporal variability and optimizing the NM segmentation of a continuous postural time-varying series. Although dynamic time warping has commonly been applied in aligning time-series data, it does not satisfy the triangle inequality[43]. Thus, we used the improved DTAK method to

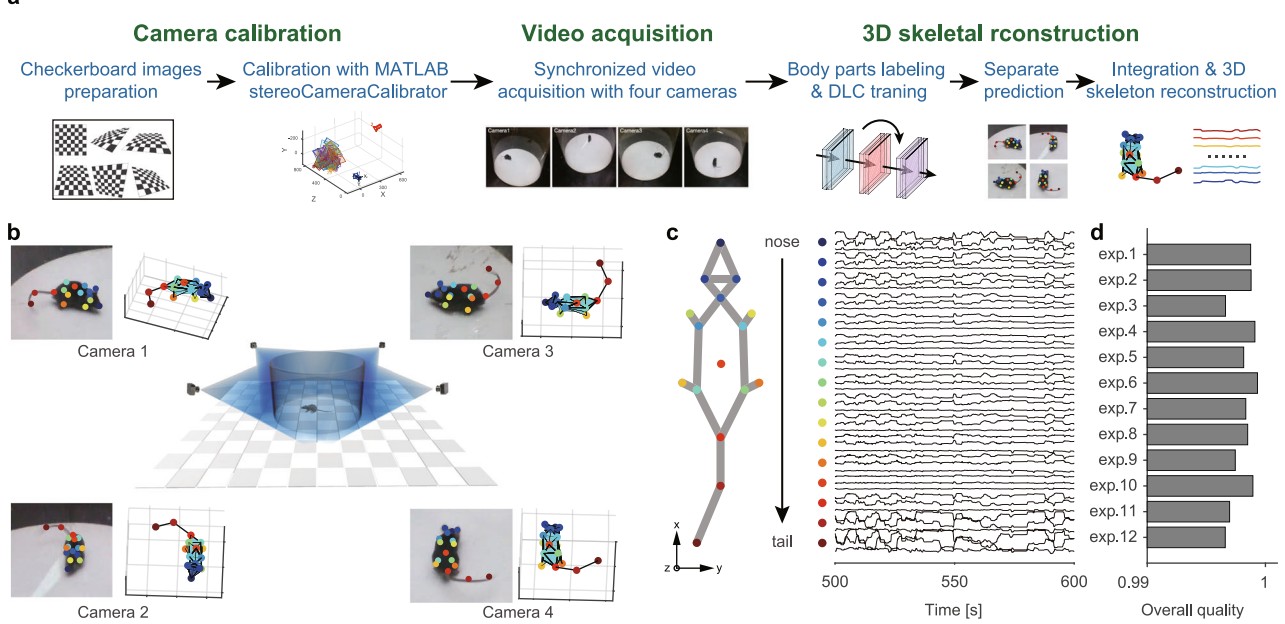

**Fig. 2 Collecting animal behavior trajectories via a 3D motion-capture system. a** Pipeline of 3D animal skeletal reconstruction. **b** Center, schematic diagram of recording animal behavior with four synchronized cameras; corners, frames captured by the cameras with the DLC labels (left) and the corresponding reconstructed skeletons (right). **c** Left: 16 key body parts include the nose, left ear, right ear, neck, left front limb, right front limb, left hind limb, right hind limb, left front claw, right front claw, left hind claw, right hind claw, back, root tail, middle tail, and tip tail. Right: representative mouse body tracking trace data collected over 100 s showing 48 data vectors obtained by DLC for each body part (indicated with a color-coded dot) encoded by x, y, and z coordinates. For visualization purposes, mean normalization is applied to each trace. **d** 3D reconstruction quality assessment: 1—best quality, 0—worst quality. The quality of the data obtained from the 12 mice averaged at 0.9981 ± 0.001. Source data are provided as a Source Data file.

measure the similarity between time sequences and construct an energy equation (objective function) for optimization. The relationship between each pair of segments was calculated with the kernel similarity matrix $K$ (Fig. 3c). DTAK was used to compute the normalized similarity value of $K$ and generate the paired-wise segment kernel matrix $T$ (Fig. 3d).

Because dynamic temporal segmentation is a non-convex optimization problem whose solution is very sensitive to initial conditions, this approach begins with a coarse segmentation process based on the spectral clustering method, which combines the kernel $k$-means clustering algorithms. To define the timescale of segmentation, the algorithm sets the maximum and minimum lengths $[w_{min}, w_{max}]$ to constrain the length of the behavioral component. For the optimization process, a dynamic programming (DP)-based algorithm is employed to perform coordinate descent and minimize energy. For each iteration, the algorithm updates the segmentation boundary and segment kernel matrix until the decomposition reaches the optimal value (Fig. 3e, f). The final segment kernel matrix represents the optimal spatial relationship between these NM segments, which can be further mapped into its feature space in tandem with dimensionality reduction (DR).

We demonstrate the pipeline of this two-stage behavior decomposition (Fig. 3h) in a representative 300-s sample of mouse skeletal data. The raw skeletal traces were segmented into NM slices of an average duration of $0.89 \pm 0.29$ s. In these segments, a few long-lasting movements occurred continuously, while most others were intermittent (Fig. 3g). The trajectories of these movement slices can reflect the actual kinematics of the behaving animal. For instance, when the animal is immobile, all of its body parts are still; when the animal is walking, its limbs show rapid periodic oscillations. Consistent with our observations, the movements corresponding to the other two opposite NMs, left and right turning, tended to follow opposite trajectories.

These preliminary results demonstrated that DTAK could be used for the decomposition and mapping of NMs.

**Mapping mouse movements with low-dimensional embeddings and unsupervised clustering**. We validated our framework in a single-session experiment with free-moving mouse behavioral data collected with the 3D motion-capture system. First, the two-stage behavioral decomposition strategy decomposed the 15-min experimental data into 936 NM bouts (Supplementary Movie M2). A 936 × 936 segment kernel matrix was then constructed using the DTAK metric. This segment kernel matrix could flexibly represent the relationship and provide insight into the relationships between each behavioral component sequence in their feature space. However, since the 936-D matrix cannot provide an informative visualization of behavioral structure, it is necessary to perform DR on this data. Various DR algorithms have been designed either to preserve the global representation of original data or to focus on local neighborhoods for recognition or clustering[44,45]. Thus, in animal behavior quantification, we face a trade-off between discretizing behavior to provide a more quantitative analysis and maintaining a global representation of behavior to characterize the potential manifolds of neural-behavioral relationships[46]. Therefore, we first evaluated the commonly used DR algorithms from the standpoints of preserving either the global or the local structure. The evaluation results show that UMAP can balance both aspects for our data (Supplementary Fig. 8) and provide 2D embeddings of these NM segments. In addition, in our parallel feature fusion framework, the factor of an animal's interaction with the environment—i.e., velocity—is considered an independent dimension. Together with 2D NM embedding, they construct a spatiotemporal representation of movements (Fig. 4a).

We used an unsupervised clustering algorithm to investigate the behavior's spatiotemporal representation and identify the

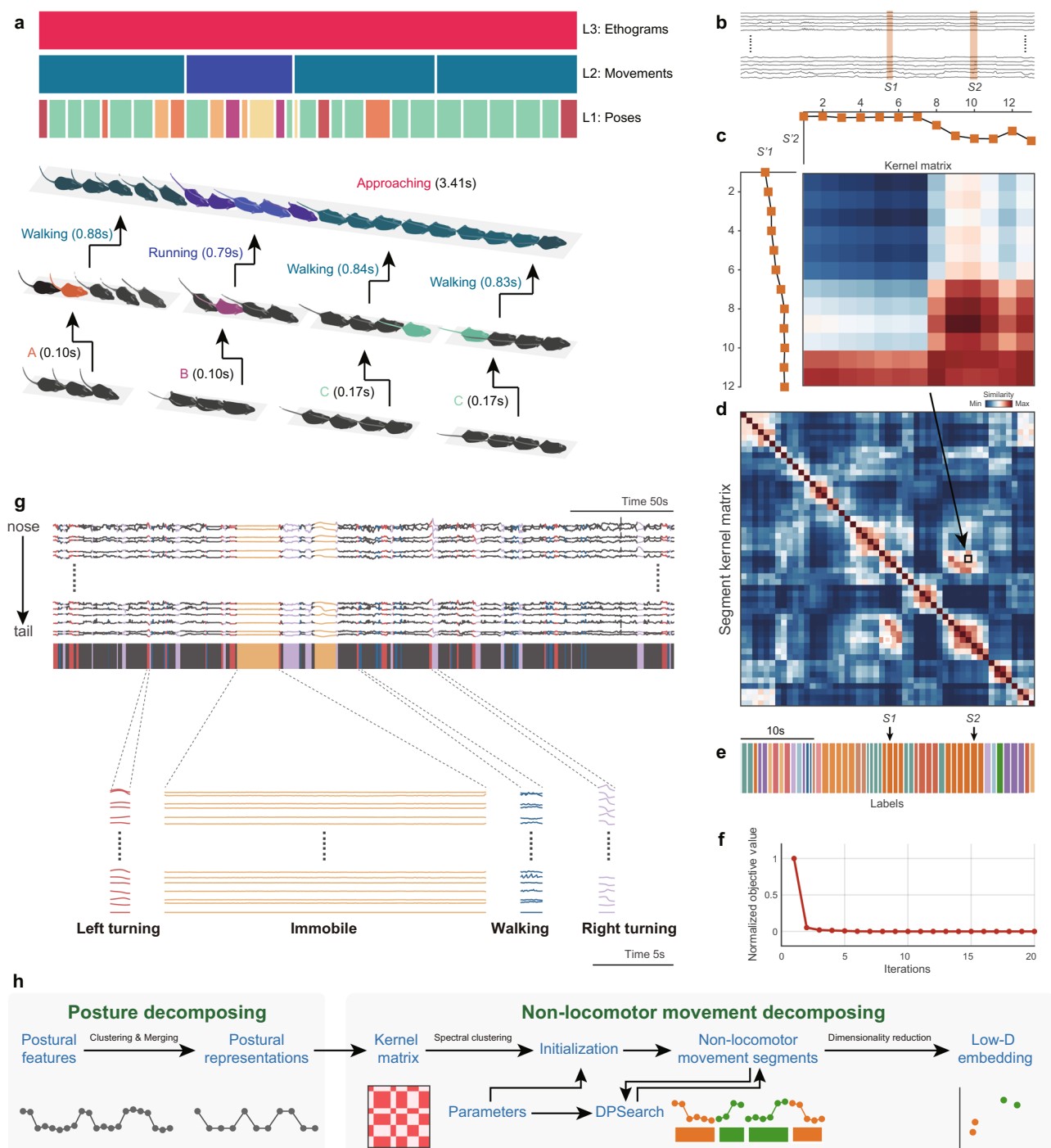

**Fig. 3 Dynamic temporal decomposition of multi-scale hierarchical behavior. a** Illustration of the three-layer bottom-up architecture for behavior. Top: The color-coded bars indicate the types of behavior components in the corresponding time period at that layer; each upper layer component is composed of the sequence of the lower layer. The instance of "approaching" is at the ethogram level which is composed of three movement level sequences, and each movement sequence includes a set of postural representations. **b** Representative animal postural trajectories (black traces) with two selected similar NM segments *S1* and *S2* (orange bars masked). **c** Discrete postural sequences *S'1* (12 points) and *S'2* (13 points) were decomposed from *S1* and *S2* and used to calculate their similarity kernel matrix *K*. **d** Segment kernel matrix *T* calculated with DTAK. Each pixel on the matrix represents the normalized similarity value of the *K* for a pair of segments at the *i*th row and the *j*th column (e.g., the pixel in the black box indicates the final similarity of *S1* and *S2*). **e** NM segments decomposed from the postural trajectories shown in **b** and their color-coded labels. Segments with the same color indicate that they belong to the same types due to their higher similarity. **f** Optimization process of dynamic temporal decomposition. Objective value (OV) error decreases with each iteration until the termination condition is reached (maximum number of iterations or OV converges). **g** Top, representative 300-s skeletal traces, where the trace slices highlighted in colors corresponding to the four types of typical NMs (left turn, immobile, walk, right turn). Bottom, magnification of representative traces of these four movement types. **h** Workflow of the two-stage behavioral decomposition. DPsearch dynamic programming search.

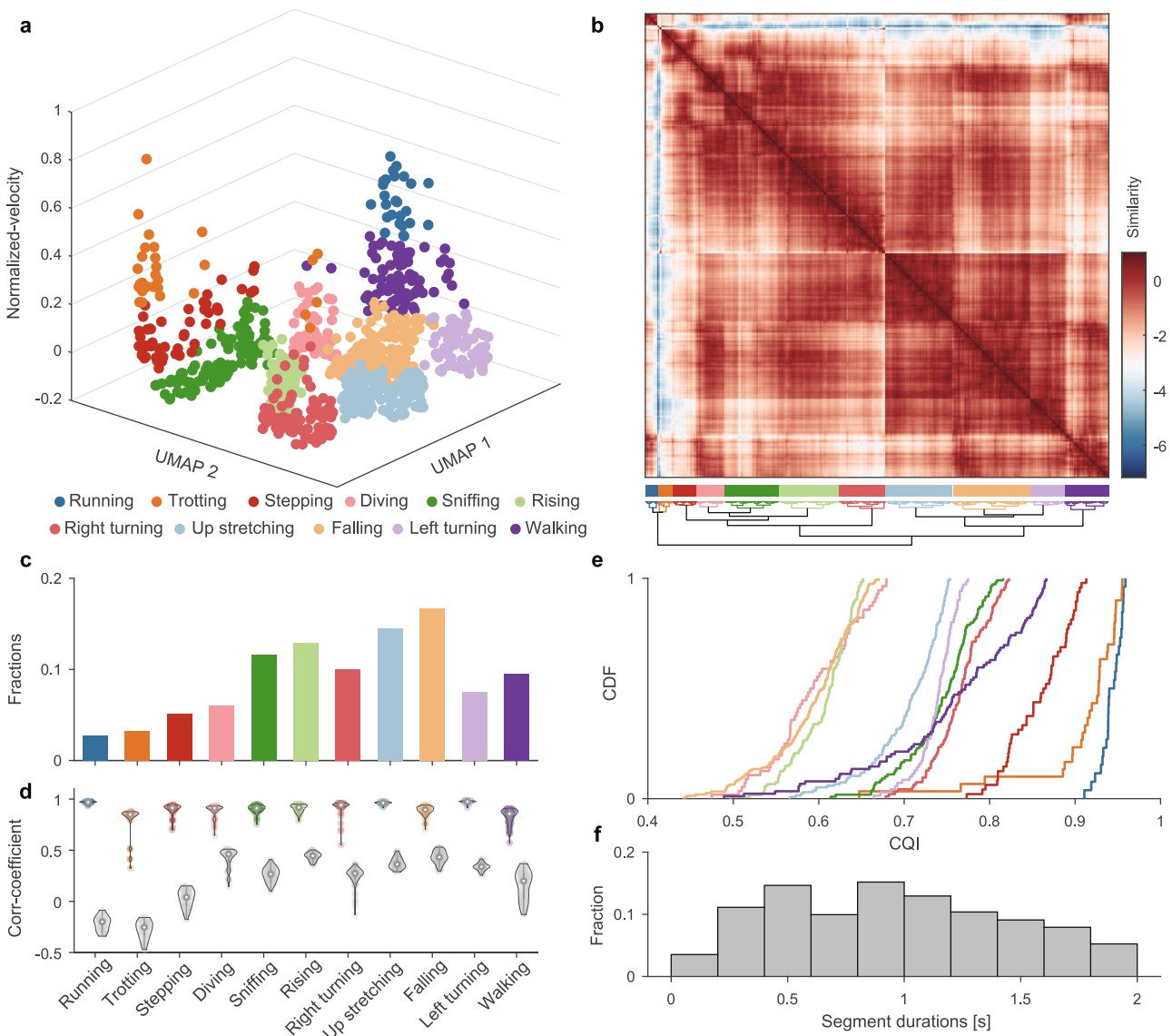

**Fig. 4 Identify movement phenotypes on single experimental data. a** Spatiotemporal feature space of behavioral components. Each dot on the 3D scatter plot represents a movement bout ($n = 935$ bouts). The 11 different colors indicate the corresponding to 11 movement types. **b** Upper, recalculated paired-wise similarity matrix, and they were rearranged with a dendrogram (lower). Each pixel on the matrix represents the normalized similarity value of a pair of movement bouts at the $i$th row and the $j$th column. The color-coded bars indicate the labels of clustered movement (middle). **c** Fractions of movement bouts number. For each subject, the behavior fractions are defined as the bouts number of each behavioral phenotype divide by the total number of behavior bouts the animal occurred during the experiment. **d** Intra-CC (color-coded) and inter-CC (gray dots) of each movement group. The dots on each violin plot represents their intra-CC or inter-CC, and dots number in a pair of violin plot in each group are the same (Intra-CC: 0.91 ± 0.07; Inter-CC: 0.29 ± 0.19). **e** Cumulative Distribution Function (CDF) of CQI of the movement clusters. The clusters represented by the curves on the right side have better clustering qualities, and their corresponding movements are more stereotyped. **f** The histogram of the duration of all movements (0.963 ± 0.497 s). CC correlation coefficient, CDF cumulative distribution function, CQI Clustering Quality Index. Source data are provided as a Source Data file.

movement phenotypes. Most unsupervised clustering require a pre-specified number of clusters, and the number chosen can be data-driven or refer to the context of the practical biological problem[47]. In the single experimental data shown in Fig. 4a, the data-driven Bayesian Information Criterion[48] in the R package *mclust* was adopted to determine that the optimal cluster number was 11 (Supplementary Fig. 10). We then recalculated the similarity matrices in the new feature space (Fig. 4b) and aggregated them using a hierarchical clustering method. Finally, we cut the original video into clips of 0.963 ± 0.497 s (Fig. 4f) and manually labeled them according to the behavior of the rodents in the clip: running, trotting, stepping, diving, sniffing, rising, right turning, up stretching, falling, left turning, and walking

(Supplementary Table 1). The locomotion types of running, trotting, stepping, and walking accounted for 20.6% of the total activities, indicating that animals spent most of the time in the NM stage (Fig. 4c).

Although we phenotyped all the clips of the entire video, it was difficult to label the behaviors of the rodents with only 11 definitions. Further, there are various heterogeneous transition stages between bouts of stereotyped movements[20,31,49]. Therefore, we evaluated them by calculating the intra-cluster and inter-cluster correlation coefficients (intra-CC and inter-CC, respectively; Figs. 4d and 5b). Our results showed that running, up stretching, and left turning have higher intra-CC and lower inter-CC, while walking and sniffing have both higher intra-CC and

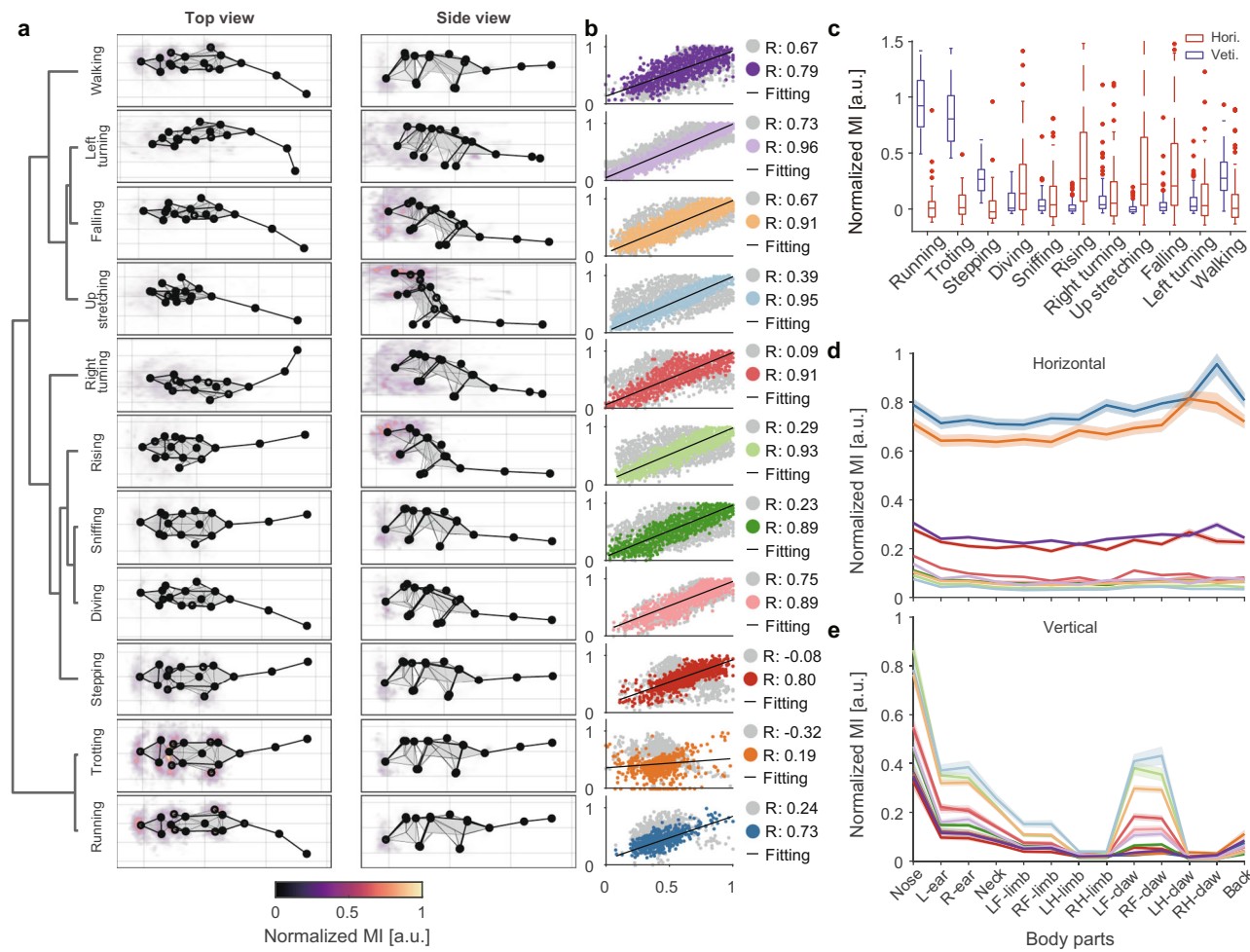

**Fig. 5 Visualization and quantification of behavioral kinematics. a** Average skeleton of all frames within each movement phenotype. The skeletons are shown with solid lines and calculated by averaging poses of body parts across time. The heatmaps overlaid on the average skeleton are the distribution and movement intensity (MI; see Supplementary Information for further details) corresponding to each movement phenotype. The MIs are scaled with 0–1 normalization method and shown in arbitrary unit (a.u.). **b** Correlation and linear regression plot of movement phenotypes. The horizontal axis represents the target, and the vertical axis represents the reference (see Supplementary Information for further details). The color-coded and gray dots correspond to the intra- and inter-cluster correlation coefficients, respectively. **c** The comparison of MI between different movement phenotypes. Each movement segment has two MI components (red boxes, horizontal; blue boxes, vertical). The boxes' values for each group contain the MIs of $n$ behavioral modules ($n$ is the number of behavioral modules of each group). In box plots, the lower and upper edges of the box are the 25th and 75th percentiles of the MIs, the central marks indicate the median, the whiskers extend to the most extreme data points not considered outliers, and the outliers are plotted individually using the dot symbol. **d**, **e** Horizontal and vertical MI of each body part in different movement phenotypes. The values on each line are the MIs of all behavior modules corresponding to the phenotype, shown by body parts separately and presented as mean ± standard deviation (SD). Source data are provided as a Source Data file.

higher inter-CC. This is because walking and sniffing co-occur with other movements[13], such as diving and turning, respectively. Finally, to evaluate the overall clustering quality, we integrated these two parameters and defined the Clustering Quality Index (CQI, Fig. 4e), which helped to determine the stereotyped/non-stereotyped movements.

**Kinematic validation of mouse behavioral phenotypes**. DTAK is an abstract extraction of animal motions that aims to simplify the complex temporal dynamics of behavior. Hence, we further elucidated whether the spatial kinematics of the original postural time-series of the behavioral phenotypes (e.g., running, rearing, sniffing, turning) identified with this framework were homogeneous. Manually inspecting the position, moving, bending, and other characteristics of the mouse limbs and trunk in the video clips of each phenotype group (Supplementary Movie M3), we

found reliable homogeneity for clips with high CQIs (>0.75). To provide a kinematic validation of the identified behavioral phenotypes from the perspectives of visualization and quantification, we first visualized the average skeleton, which was averaged over all frames in each movement cluster (Fig. 5a). While some movements could be clearly recognized (e.g., left and right turning, and up stretching), the differences between movements with similar postures (running, trotting, walking, etc.) were not. The detailed kinematic parameters, especially the velocity of each body part, could provide greater sensitive differences than the unclear visually based assessments[50]. Therefore, we defined movement intensity (MI) as a metric for characterizing the kinematics of each body part in each behavioral phenotype (see Supplementary Information for further details). MI is related to velocity, and it contains both horizontal and vertical components. The data show that the horizontal MI components of running and trotting are the highest, followed by stepping and walking.

Vertical MI components (e.g., up stretching, rising, and falling) feature richer details; we attribute their high overall vertical MI to the movement of the nose and front claws (Fig. 5a, c–e). This approach of creating portraits for each type of movement provides further support for the efficacy of our framework in the decomposition of animal behavior. The dendrogram of the movements (Fig. 5a) revealed that similar movements were arranged closely, such as running and trotting. Interestingly, falling and left turning were on close clades. Review of the video clips of these two groups demonstrated that 37.18% of the movements in this group occurred simultaneously with left turning (28.85% for right turning). A similar phenomenon occurred in the clades of diving and sniffing due to the co-occurrence of these behaviors. The correlation and linear regression analysis of these two pairs of clades showed that both intra-CC and inter-CC were relatively high (Fig. 5b), suggesting several concomitant descriptions of animal behavior. These clustering results occurred because these movements show more characteristics of the current class.

**Identification of the behavioral signatures of the mouse disease model**. Animal disease models play an increasingly critical role in expanding understanding of the mechanisms of human diseases and novel therapeutic development[51–53]. Behavioral phenotyping provides a noninvasive approach to the assessment of neuropsychiatric disorders in animal models. By only evaluating spontaneous behavior without any induced conditions, we demonstrate the usability and unbiased character of our framework for animal phenotyping. We collected data from 20 mice (Fig. 6a and Supplementary Fig. 9h, i, $n_{KO} = 10$, $n_{WT} = 10$) with our 3D motion-capture system and subjected them to routine velocity and anxiety index analyses (Fig. 6b–e). In agreement with prior research, we found a significant difference between the average velocities of the two groups.

We clustered the behavioral components of the 20 animals and obtained 41 behavioral phenotypes (Fig. 6f and Supplementary Fig. 10). Compared with the single-session experiment, the group analysis revealed diverse behavioral types. We found that *Shank3B* knockout (KO, *Shank3B*$^{-/-}$) mice spent a significantly higher proportion of their time engaging in four of the movements (Fig. 6g and Supplementary Table 3). By manually reviewing the video clips of these four types, we annotated the 38th movement (M38 in Fig. 6g) as hunching; we also found that three of the movements were very similar (closely arranged on the behavioral dendrogram, Fig. 6g). Therefore, we grouped them and annotated them as self-grooming. In previous studies[54–56], self-grooming has been widely reported in *Shank3B*$^{-/-}$ mice. This is partly attributable to self-grooming being a long-lasting movement (4.48 ± 7.84 s, mean ± standard deviation [SD]) and thus easily recognized by human observation or software (Fig. 6i). Interestingly, although hunching has only previously been reported in a few related studies[57–59], our framework frequently detected hunching movements in KO mice. This novel finding can be attributed to the duration of a single continuous hunching movement being too short to be noticed (1.29 ± 1.00 s, mean ± SD) as well as to the similarity between the kinematics of hunching and rearing (M31). We proved that these two types of movements belong to distinct behavioral phenotypes. Specifically, during hunching, mice maintain an arcuate spine angle, while rearing is characterized by a stronger, wider range of necks and head motions (Fig. 6j–n). This ability to identify short-term and fine behavioral modules is one of the advantages of our framework. Besides the four phenotypes that KO mice preferred more than the WT mice did, the KO mice also showed four additional deficit behavioral phenotypes, namely stepping (M5), walking (M14),

and two types of rising (M21 and M22). This result indicates that the locomotion intensity and vertical movement of KO mice were lower than those of WT mice. The locomotion result is consistent with the average velocity comparison shown in Fig. 6b.

Finally, we demonstrated that by modeling the time spent of multi-behavioral parameters, our framework could identify the animal types. We used UMAP to perform DR of the 41-dimensional behavioral proportion data of all movement types. As expected, the two genotypes of animals were well separated in the low-dimensional space (Fig. 6h), even though there were large amounts of baseline movements with no significant difference. We defined these two types as "autistic-like behavior space." Recent reviews suggest that most previous methods[60,61], which usually only consider a few behavioral parameters and may lose critical insights, have been challenged in the animal phenotypes' identification. Hence, these findings indicate the potential advantages of our framework to automatically identify disease models.

## Discussion

Inspired by the natural structure of animal behavior, the current study presents a framework for discovering quantifiable behavioral modules from high-dimensional postural time-series by combining dynamic temporal decomposition and unsupervised clustering. Behavior decomposition adopts a parallel, two-stage approach to extract animal motion features in accordance with the natural structure of animal behavior. We used DTAK to measure the similarity between behavioral modules and applied further low-dimensionality embedding to represent the behavior's underlying feature space. The unsupervised clustering identified behavioral phenotypes from the feature space and helped to automatically assess the behavioral experiment data. In addition, the clustering step could quickly generate large amounts of distinct unlabeled behavior groups. By manually assigning annotations to each group, our framework will potentially facilitate semi-supervised behavior recognition.

Our framework has two main advantages. First, our approach of tracking multiple body parts and acquiring 3D reconstruction data achieves better performance than similar recently reported rodent behavioral recognition frameworks[14,62]. The multi-view motion-capture system can avoid animal body occlusion and view-angle bias and estimate the pose optimally by flexibly selecting the view to use according to the tracking reliabilities of the different views. We also confirmed the necessity of using multi-view cameras in complex experimental scenes, whereas in the simple experimental scenes, only three or even two cameras were needed (Supplementary Fig. 4). Currently, we are working on conducting a comprehensive comparison between our framework and traditional behavior analysis approach for evaluating the anxiety-like mouse models' behavior. The preliminary results showed that compared with the single-camera solution, our approach could significantly identify the behavioral differences of the anxiety-like mice, whereas the traditional method could not detect the significance. More importantly, our behavior decomposition framework emphasizes the extraction of the temporal dynamics of movements. Without making model assumptions, similar movements with various time durations and temporal variability can be efficiently represented by the self-similarity matrix. We proved that this similarity matrix is a reliable objective metric by evaluating the consistency of clustered behavior phenotypes. We further performed DR to visualize the behavioral map, which facilitates exploring the evolution of movement sequences of higher-order behavior and behavioral state transition caused by neural activity. For example, to study animal circadian rhythms, previous researchers have used behavioral

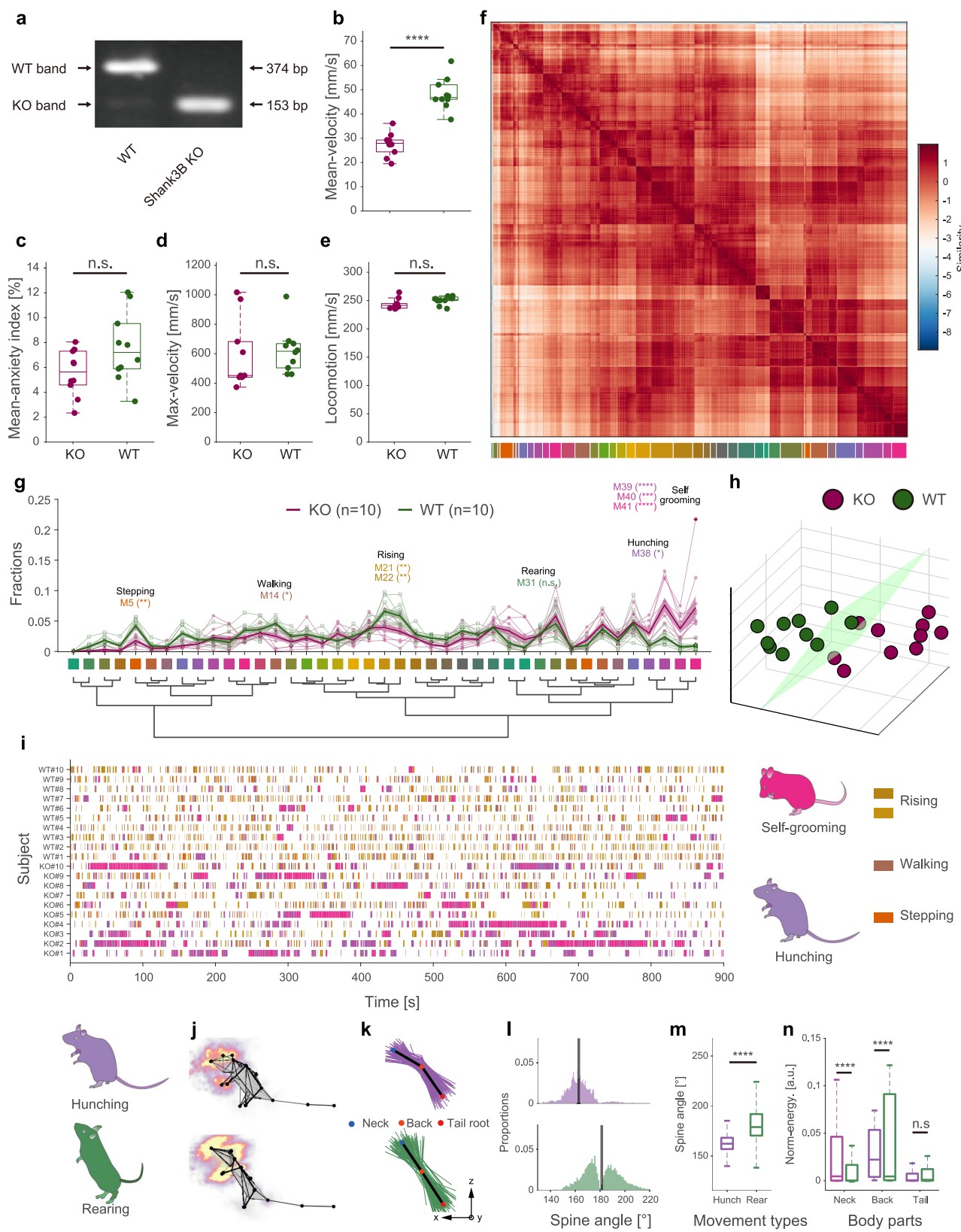

recording approaches to characterize different brain states[63–65]. We used our framework to perform a continuous 24-h behavioral recording, and the preliminary analysis proved that our framework could provide more comprehensive behavioral parameters and detailed quantification of behavior states (Supplementary Fig. 13). In addition, innate defensive behavior is considered to consist of three specific movement phases[37,66], but data supporting this idea are lacking. Hence, our future work will focus on modeling the transition patterns of innate behavior based on the behavioral map.

Comprehensive and unbiased behavioral phenotyping is becoming a powerful approach to the study of behavioral

**Fig. 6 Spontaneous behavior analysis reveals autistic-like behaviors in *Shank3B* knockout mice. a** PCR genotyping for *Shank3B*$^{+/+}$ (wild type, WT), *Shank3B*$^{-/-}$ (*Shank3B* knockout, KO) mice. The full scans of all the sample can be found in Supplementary Fig. 9. **b–e** Box plots of mean velocity, mean anxiety index, maximum velocity, and locomotion of the two groups of animals (purple: KO, $n = 10$, green: WT, $n = 10$; statistics: two-sided Mann–Whitney test for maximum velocity; two-sided unpaired *T*-test for others, ****$P = 9.0549 \times 10^{-8}$, $t = 8171$, DF = 18), values are represented as mean ± SD. **f** Top: recalculated paired-wise similarity matrix. The movement bouts of all 20 mice involved were grouped ($n = 16607$) and rearranged in a dendrogram (**g**). Each pixel on the matrix represents the normalized similarity value of a pair of movement bouts at the *i*th row and the *j*th column. The color-coded bars (41 clusters) indicate the movements being clustered (bottom). **g** Comparison of the fraction of movement types between KO mice and WT mice. The bold traces and shadows indicate the mean ± s.e.m. Fractions of each group and light color traces are the fractions of all 20 mice (purple, KO, $n = 10$; green, WT, $n = 10$). Middle color-coded labels and dendrogram indicate the movement types. Eight movements have significant differences between the two groups, and the fractions of the four movements that KO mice prefer are hunching (M38, KO = 3.00 ± 0.56%, WT = 0.94 ± 0.15%) and self-grooming groups (M39, K = 7.65 ± 1.21%, W = 2.34 ± 0.33%; M40, K = 3.73 ± 0.72%, W = 0.75 ± 0.19%; M41, K = 7.23 ± 1.88%, W = 0.90 ± 0.18%). Statistics: two-way ANOVA followed by Holm–Sidak post hoc multiple comparisons test, **M5, $P = 0.0065$; *M14, $P = 0.0392$; **M21, $P = 0.0012$; **M22, $P = 0.0030$; *M38, $P = 0.0456$; ****M39, $P < 0.0001$; ***M40, $P = 0.0001$; ****M41, $P < 0.0001$. **h** Low-dimensional representation of the two animal groups (purple, KO, $n = 10$; green, WT, $n = 10$). The 20 dots in 3D space were dimensionally reduced from 41-dimensional movement fractions, and they are well separated. **i** Ethograms of the eight significant movements. **j–n** Kinematic comparison of rearing and hunching (upper row refers to hunching; lower row refers to rearing). **j** Average skeletons of all frames and normalized moving intensity (side view) of rearing and hunching. **k** Spine lines (the lines connecting the neck, back, and tail root) extracted from all frames (rearing, 16,834 frames; hunching, 10,037 frames) in movement types. For visualization purposes, only 1% of spine lines are shown in the figure (rearing, 168/16,834; hunching, 100/10,037). Black lines refer to the averaged spine line of the hunching and rearing phenotypes; **l** histograms of the spine angles (angle between three body parts). During rearing, the spine angles of the animals swing, and the average spine angle is straight (181.34 ± 15.63°). By contrast, the spine angles of the rodents during hunching are consistently arcuate (162.88 ± 10.08°). **m**, **n** Box plot of spine angles of the two movement types. **n** Box plot of normalized MI of the three body parts involved. Statistics for **m**, **n**: two-sided Mann–Whitney test. ****$P < 0.0001$. In box plots, the lower and upper edges of the box are the 25th and 75th percentiles of the values, the central marks indicate the median, and the lower and upper whiskers are the minima and maxima values. Source data are provided as a Source Data file.

abnormalities in animal models of neuropsychiatric disorders. In this study, we demonstrate its application to the monitoring of *Shank3B* mutant mice that show autistic-like behaviors. Our framework helped to reveal that *Shank3B*$^{-/-}$ engage in eight types of spontaneous behaviors significantly more often than WT mice; while grooming has been extensively observed in murine models of restricted, repetitive behavior, short-term hunching behavior has not. Previous studies[54,55] mentioned that the rearing behavior of *Shank3B* KO mice also differs from that of WT mice; however, because hunching is kinematically similar to rearing, it is difficult to distinguish these two types by human observation or algorithms. Our 3D and sub-second methods will help to identify new behavioral biomarkers and advance understanding of the neural circuit mechanisms underlying behavioral changes caused by genetic mutations. Moreover, we further investigated the differences in the behavior patterns of *Shank3B* KO and WT mice at the group level. In addition to the data that had already been analyzed (collected under the condition: male mice, 5–6 weeks, white light, and circular open field), we extended the group behavioral pattern analysis to include data collected under different conditions (i.e., different experimental apparatus, lighting, age, and sex; Supplementary Table 2). We calculated the cross-correlation coefficient matrix (CCCM) of all samples based on the movement fractions and used principal component analysis to extract the main variance factors of the CCCM (Supplementary Fig. 12a, b). We found that when only a single condition was changed for male mice, there was no significant difference in population behavior patterns in mice with the same genotype (Supplementary Fig. 12c). We also found that although some female KO mice had a weak tendency for autistic-like behavior, there was no significant difference between 5–6 week male and female KO mice at the group level when tested under the white-light circular open-field condition (Supplementary Fig. 12c, d). Finally, we compared the behavior patterns when all conditions were the same except for the genotypes. The results showed that only the female group showed no significant difference between KO and WT genotypes, while significant differences in behavioral patterns were found between KO and WT male mice under all other conditions. These findings are consistent with previous reports that *Shank3B* KO male mice display more severe

impairments than females do in motor coordination[67]. Accordingly, the behavior phenotyping on mouse disease model can be generalized to large animals such as non-human primates, dogs, and pigs which recently emerged as valuable models for studying neurological dysfunctions[52,53]. Our general-purpose framework further benefits from the significant advantage of being able to capture and analyze large animal movements, which have more complex 3D characteristics and temporal dynamics.

The dynamic, high-dimensional, and multi-scale characteristics of behavior can be attributed to similar properties of the nervous system that produces it. While the most advanced large-scale neuroimaging and high spatiotemporal resolution electrophysiological techniques allow researchers to elucidate the details of the firing timing of all neurons and neurofunctional connections at all scales, they cannot inform the mapping of the neural-behavioral relationship without quantifying behavior at the corresponding level. In other words, to understand the encoding/decoding relationship rules of the neural activity generating behavior and behavior's neural representation, synchronization of large population activities and accurate measurement and identification of naturalistic, complex behavior are required. In the future, we will focus on combining our framework with free-moving two-photon microscopy and electrophysiological recording to link the neural activity patterns and functional brain connections with the cross-scale behavioral dynamics and timing patterns. Therefore, with further technical optimization and the open source of a large sample, well-annotated disease model behavior database open source, our framework may contribute to resolving the relationships between complex neural circuitry and behavior, as well as to revealing the mechanisms of sensorimotor processing.

Lastly, we would like to discuss the limitations of our framework. When extending our framework to social behavior analysis, such as the analysis of mating, social hierarchy, predation, and defense behaviors, it is challenging to track multiple, visually indistinguishable (markerless) animals without identity-swapping errors (Supplementary Movies M6 and M7). Alternative methods mainly focus on tracking and identifying social behaviors at the population level, which only requires the identification of features unrelated to the animals' identities such as the positional

differences between animals' body parts[68]. However, this approach is limited to specific behaviors and does not apply to interaction behaviors between social subjects of unequal status. Recent cutting-edge toolboxes such as DLC for multi-animal pose estimation[17], SLEAP[69], and AlphaTracker[70] have addressed the multi-animal tracking problem, but once animals with similar appearances are touching or even body-occluded, the inaccurate pose estimation of these toolboxes leads to off-tracking and identity-swapping errors. This is because when estimating multiple body parts of several animals in a single frame, the combination of the poses of these animals is more complex and diverse, and identity-swapping in different views may happen at different times. Our 3D multi-view motion-capture system promises to solve this problem by effectively reducing body-occlusion probability. As a next step, we are considering using computer vision technology (e.g., point cloud reconstruction) to fuse images from multiple views, then segment each animal's body, and estimate the body parts based on the reconstructed 3D animal. Solving these problems will extend the applicability of our framework to the benefit of the animal behavioral research community.

## Methods

**Apparatus**. The multi-view video capture device is shown in Supplementary Fig. 1a and Supplementary Fig. 2a, b. Mice were allowed to walk freely in a circular open field made of a transparent acrylic wall and white plastic ground, with a base diameter of 50 cm and a height of 50 cm[71]. The circular open field was placed at the center of a $90 \times 90 \times 75$ cm³ movable, stainless-steel support framework. A black, thick, dull-polished rubber mat was paved between the circular open field and steel shelf to avoid light reflection. Four Intel RealSense D435 cameras were mounted orthogonally on the four supporting pillars of the shelf[72]. Images were simultaneously recorded at 30 frames per second by a PCI-E USB-3.0 data acquisition card and the pyrealsense2 Python camera interface package. On the top of the shelf, a 56-inch TV was placed horizontally, facing down, to provide uniform and stable white background light. The cameras and TV were connected to a high-performance computer (i7-9700K, 16G RAM) equipped with a 1-terabyte SSD and 12-terabyte HDD as a platform for the software and hardware required for image acquisition.

**Animals, behavioral experiments, and behavioral data collection**. Adult (5–6 weeks old or 11–13 weeks old) *Shank3B* knockout (KO; *Shank3B*$^{-/-}$) and wild-type (WT; *Shank3B*$^{+/+}$) mice, on a C57BL/6J genetic background, were used in the behavioral experiments (Supplementary Fig. 9 and Supplementary Table 2). *Shank3B*$^{-/-}$ mice were obtained from the Jackson Laboratory (Jax No. 017688) and were described previously[54]. The mice were housed at 4–6 mice per cage under a 12-h light–dark cycle at 22–25 °C with 40–70% humidity, and were allowed to access water and food ad libitum. All husbandry and experimental procedures in this study were approved by Animal Care and Use Committees at the Shenzhen Institute of Advanced Technology (SIAT), Chinese Academy of Sciences (CAS). All behavioral experiments were performed and analyzed with experimenters blinded to genotypes.

We designed two behavioral experiments. In the first, we collected behavioral data under different conditions in terms of genotype, age, sex, experimental apparatus, and lighting conditions, yielding the following 10 groups (see Supplementary Fig. 12 and Supplementary Table 2): (1) KO1: *Shank3B*$^{-/-}$, 5–6 weeks old, male, circular open field, white light; (2) KO2: *Shank3B*$^{-/-}$, 5–6 weeks old, male, square open field (50 × 50 cm), white light; (3) KO3: *Shank3B*$^{-/-}$, 5–6 weeks old, male, circular open field, infrared light; (4) KO4: g*Shank3B*$^{-/-}$, 11–13 weeks old, male, circular open field, white light; (5) KO5: *Shank3B*$^{-/-}$, 5–6 weeks, female, circular open field, white light; (6) WT1: *Shank3B*$^{+/+}$, 5–6 weeks old, male, circular open field, white light; (7) WT2: *Shank3B*$^{+/+}$, 5–6 weeks old, male, square open field, white light; (8) WT3: *Shank3B*$^{+/+}$, 5–6 weeks old, male, circular open field, infrared light; (9) WT4: *Shank3B*$^{+/+}$, 11–13 weeks, male, circular open field, white light; and (10) WT5: *Shank3B*$^{+/+}$, 5–6 weeks old, female, circular open field, white light. Each mouse in each group was only used once. Mice were first acclimatized to the field for 10 min and then recorded for another 15 min. We wrote Python and OpenCV programs to obtain and record the videos of mice. The frame rate was set to 30 fps, and the frame size was set to 640 × 480.

In the second behavioral experiment, we assessed mouse behavior in the circular open field for 24 h (related to Supplementary Fig. 13). In this experiment, the circular open field was covered by wood chip for padding, and mice had access to water and regular food (chow diet). Male, 13-week-old mice with a C57BL/6J genetic background were used. In order to change the light conditions but maintain the circadian rhythms of the mice, we used an infrared light as the background light and set the cameras to the infrared mode. This experiment was started at 20:20 p.m. We first turned off the light, leaving it off until 7:00 a.m. the next day,

and then turned on the white light until 19:00 p.m. Lastly, we turned off the light until 20:20 p.m. and finished the 24-h behavioral recording. Detailed information for the mice and experimental conditions used are presented in Supplementary Table 2.

**Evaluation of 3D reconstruction quality obtained by different camera settings**. To test the limit of reducing the number of cameras, we performed a detailed analysis of the precision quantification by different camera settings (Supplementary Fig. 4). For each camera, there was no significant difference in the likelihoods of 2D pose estimation, indicating that the position of the camera had no significant impact on the estimation result. In the 3D reconstruction procedure, it was enough to apply only two calibrated cameras for the acquisition of 3D body points. As such, the best two points with the highest likelihoods from all four cameras were selected to be reconstructed in 3D space. We chose four different camera groupings (2C180, 2C90, 3C, 4C) to confirm reconstruction accuracy (Fig. 2b). The likelihoods of 2C180 and 2C90 were not significantly different, indicating that, in the case of using two cameras, the position differences do not affect the basic 3D reconstruction of the mouse. The likelihoods of 3C and 4C were also not significantly different, suggesting that three cameras could basically reach the precision requirement of four cameras in this case. The likelihoods of both 3C and 4C were significantly higher than those of 2C180 and 2C90 (one-sided Kruskal–Wallis test followed by Dunn's multiple comparisons test, $p = 0.1339$, $n = 16$). This result is consistent with the results of multiple previous studies, showing that the number of cameras is positively correlated with the accuracy of pose tracking[73,74]. From this point of view, the minimum number of cameras was determined to be two. However, for less occlusion, our results and those of previous studies suggest increasing the number of cameras when focusing on different animals.

Further, we calculated the variances in 3D behavioral trajectories obtained by different camera settings to test the limit of reducing the number of cameras. Variance is a representation of data information, including objective information and noise information. When the variance is converged with more cameras, we determined the least number of cameras to reduce the cost. Statistical analysis showed that 2C180 was significantly higher than 2C90, and that 2C90 was markedly higher than 3C. However, 3C and 4C were not significantly different. According to their proportional relations, 2C180 yielded more than 30% noise information compared with the respective noise information obtained with 2C90, while 2C90 yielded more than 50% noise information when compared with the respective information obtained with 3C and 4C. Thus, the most suitable and economical number of cameras in this experiment situation was three.

Lastly, we tested the influence of accuracy of each body part by different camera settings. We calculated the variances in accurately capturing each body part in each X, Y, and Z coordinate of 3D poses. Different camera settings yielded similar variances in the accuracy of capturing different body points. The variance in the accuracy of capturing tails in 3D poses was higher than that of other body parts. These results suggest that, for different body parts, the noise level and tracking ability of the same camera settings are different, and that tracking specific body parts requires balancing the number, quality, and grouping of cameras. For example, if the movement of the tail is the main research focus, it is important to add more cameras to improve the tracking accuracy.

**Proportion of available cameras for tracking each body part**. In the 3D reconstruction step, it is possible that not all 2D body parts captured by multi-view cameras can be used for 3D reconstruction due to tracking failure caused by occlusion (Supplementary Figs. 5 and 6). For this reason, we evaluated the accuracy of the 3D reconstruction based on the number of available cameras for each body part. A threshold of the likelihood of pose estimation was used to assess whether the tracking of specific body parts was successful or not. If the likelihood was larger than the threshold, the camera was used. The threshold was set to 0.9, and we used a video of 27,000 frames for calculating the proportion of the numbers of available cameras. The number of cameras needed is shown using scaled stack bars.

**Determination of clustering number**. In the clustering step, selecting an appropriate number of clusters is critical (Supplementary Fig. 10). However, in most cases, an appropriate number of clusters is difficult to determine because most movements have high similarity, thus blurring the boundaries of different movement clusters in the movement space. Hence, we determined the appropriate number of clusters by Bayesian Information Criterion (BIC) in "mclust" package of R language[48]. This package provides 14 different types of models to infer the best parameters, such as the number of clusters. In our tests, the best number of clusters in single behavioral experiments ranged from 10 to 20. Considering the discrimination of different behaviors while giving them suitable labels, we chose 11 as the most appropriate number of clusters in movement clustering. When clustering all the movements of 10 *Shank3B*$^{+/+}$ and 10 *Shank3B*$^{-/-}$ mice in the movement space, we used the recommended number of clusters of BIC with manual inspection, which was 41. We used 41 clusters because it was not possible to label all the clusters. Most movements were not different between the two genotypes; thus, we only labeled the different movements and some critical movements.

**Behavioral phenotypes' definition**. We defined 14 different behavioral phenotypes (Supplementary Table 1) referring to Mouse Ethogram database (www.mousebehavior.org) and ref. [57].

**Statistics**. The sample sizes of behavior tests were selected by referring to the previous related studies[54,55,67,75] and verified by power analysis[76]. All related data are included in analysis. All mice were tested once, all attempts at replication were successful, and no individual replicate was excluded. Analyses were performed using Prism 8.0 (GraphPad Software). Before hypothesis testing, data were first tested for normality by the Shapiro–Wilk normality test and for homoscedasticity by the $F$ test. For normally distributed data with homogeneous variances, parametric tests were used (Student's $t$-test for two groups, two-way ANOVA followed by Holm–Sidak post hoc test for more than two groups); otherwise, non-parametric tests were used (Mann Whitney test for two groups, two-way ANOVA with Bonferroni's post hoc test for more than two groups).

The mean velocity data of 10 $Shank3B^{+/+}$ and 10 $Shank3B^{-/-}$ mice (Fig. 5b) were normally distributed and their variances were homogeneous; thus, a two-sided unpaired $t$-test was used to compare the difference between the two groups. The mean anxiety index data of these two groups (Fig. 5b) were not normally distributed; thus, we used the two-sided Mann–Whitney test to compare the differences between groups. The movement fractions data (Fig. 5d) were normally distributed, with homogeneous variances; thus, two-way ANOVA followed by Holm–Sidak post hoc multiple comparisons test was used to compare the differences among the groups.

**Autistic-like behavior space**. The fraction of 41 cluster movements of each mouse (Fig. 6g) was considered as a feature vector of the mouse behavioral space, with the feature vector set as $\mathbf{x} = [x_1, x_2, ..., x_{41}]^T$ and the feature matrix of behavior space set as $\mathbf{X} = [\mathbf{x}_1, \mathbf{x}_2, ..., \mathbf{x}_{12}]^T$. We then used UMAP to reduce the feature dimensions of $\mathbf{X}$ from 41 to 3 dimensions, as follows:

$$\mathbf{Y} = f_{\text{UMAP}}(\mathbf{X}) \tag{1}$$

where $\mathbf{Y} = [\mathbf{y}_1, \mathbf{y}_2, \mathbf{y}_3]^T$, $\mathbf{y}_i = [y_i^1, y_i^2, ..., y_i^{41}]^T$ is a 3D feature matrix, the autistic-like behavior space, after the dimension reduction by UMAP. $f_{\text{UMAP}}(\cdot)$ included the parameters n_neighbors set to 30 and min_dist set to 0.3, which are robust enough to change across a wide range and discriminate between $Shank3B^{+/+}$ and $Shank3B^{-/-}$ mice in the autistic-like behavior space. To quantify the group differences, we fitted a linear classification model to $\mathbf{Y}$ by the fitclinear function in MATLAB by using the default parameters.

**Reporting summary**. Further information on research design is available in the Nature Research Reporting Summary linked to this article.

## Data availability

All the raw videos and 3D skeleton trajectories associated with $Shank3B^{-/-}$ mice spontaneous behavior test showing in Fig. 6 are available in the Zenodo repository[77] https://doi.org/10.5281/zenodo.4629544 and Supplementary information. Any other relevant data are available upon reasonable request. Source data are provided with this paper.

## Code availability

The codes of this framework can be accessed at https://behavioratlas.tech/ and are available on Zenodo[78]: https://doi.org/10.5281/zenodo.4626951.

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

## Acknowledgements

We thank Jianyuan Sun, Guoqiang Bi, Yu-Ting Tseng, and Yunke Wang for comments on our manuscript; Feng Wang, Zhonghua Lu, and Yu Hu providing *Shank3B−/−* mice. This work was supported in part by Key Area R&D Program of Guangdong Province (2018B030338001 to P.W., 2018B030331001 to L.W.), National Key R&D Program of China (2018YFA0701403 to P.W.), National Natural Science Foundation of China (NSFC 31500861 to P.W., NSFC 31630031 to L.W., NSFC 91732304 to L.W., NSFC 31930047 to L.W.), Chang Jiang Scholars Program (to L.W.), the International Big Science Program Cultivating Project of CAS (172644KYS820170004 to L.W.), the Strategic Priority Research Program of Chinese Academy of Science (XDB32030100 to L.W.), the Youth Innovation Promotion Association of the Chinese Academy of Sciences (2017413 to P.W.), CAS Key Laboratory of Brain Connectome and Manipulation (2019DP173024), Shenzhen Government Basic Research Grants (JCYJ20170411140807570 to P.W., JCYJ20170413164535041 to L.W.), Science, Technology and Innovation Commission of Shenzhen Municipality (JCYJ20160429185235132 to K.H.), Helmholtz-CAS joint research grant (GJHZ1508 to L.W.), Guangdong Provincial Key Laboratory of Brain Connectome and Behavior (2017B030301017 to L.W.), the Ten Thousand Talent Program (to L.W.), the Guangdong Special Support Program (to L.W.), Key Laboratory of SIAT (2019DP173024 to L.W.), Shenzhen Key Science and Technology Infrastructure Planning Project (ZDKJ20190204002 to L.W.).

## Author contributions

K.H., Y.H., P.W., S.L. and L.W. conceived, designed, and implemented the framework; P.W., S.L., K.H. and Y.H. designed and implemented the behavior decomposition algorithm; K.H., Y.H. and K.C. designed and implemented the 3D motion-capture system; Y.H., H.P., K.H., G.Z., K.C. and W.Y. collected the animal behavioral data; K.H., Y.H., K.C., P.W. and S.L. performed data analysis; K.H., Y.H., P.W., S.L., L.W. and X.L. wrote the manuscript; P.W., S.L., L.W., K.H. and Y.H. discussed the results and revised of the manuscript; and P.W. and L.W. acquired the funding.

## Competing interests

The authors declare no competing interests.
