## [Peer Review File · Nature Communications]

Reviewers' Comments:

Reviewer #1:

Remarks to the Author:

This paper presents a parallel, multi-layered framework to learn the hierarchical dynamics and generate an objective metric to map the behaviour of a mouse into the feature space. Furthermore, 3D kinematics with the low-cost multi-view motion capture system have been analysed and discussed in this paper. The authors also demonstrated that the proposed technology can identify the animal behaviours and transgenic animal disease models from the behaviour monitoring.

The major weakness of this paper is the lack of novel contribution. The proposed system mainly consists of a number of standard components developed by other researchers in the community. In addition, the bio-markers identified in this research lack convincing supportive evidence. The reviewer will further comment on these aspects in the following discussion.

1. Novelty of the research:

The authors stated that "most recent end-to-end machine learning based behaviour analysis methods focused on recognizing behavioural identities in a static way or based on limited observations". The statement was based on limited survey on the related topics. In fact, a number of research projects have addressed dynamical and continuous activities of mice, e.g.

(1) <https://www.sciencedirect.com/science/article/pii/S0165027017301139>, (2) <https://journals.plos.org/plosone/article?id=10.1371/journal.pone.0220751>.

(1.1) 3D motion capture:

The 3D motion capture system is based on the standard apparatus, camera calibration, pose estimation (Ref 17), and 3D skeletal reconstruction (Ref 41). It is reasonable to include the state of art technologies at this stage. However, the issues are, the stages of pose estimation/3D skeletal reconstruction miss the investigation on body occlusion and view-point disappearance.

(1.2) Non-locomotor movements with dynamic time alignment Kernel:

Center alignment and rotation transformation have been applied in the pre-processing stage. This process is tedious indeed. Afterwards, a temporal reduction algorithm is used to merge the adjacent similar poses, which group wrong poses into individual pose categories. The standard DTAK method is also used to measure the similarity between sequences. However, it is not clear why not use the standard DTW method in this case.

(1.3) Mapping mouse movements with low-dim embeddings and unsupervised clustering:

A standard UMAP method is used to preserve both the local and global structure of the dataset. However, it is not clear to the reader why this method must be used - lacks supporting evidence. BIC is used to model the structure but how?

(1.4) Kinematic validation of mouse behavioural phenotypes:

MI components have been constructed to describe the postural difference but 'why and how' are not explained.

2. Identification of behavioural signatures of mouse disease models:

Two groups of mice have been used to investigate possible bio-markers. 6 KO and 6 WT mice may suggest something but more substantial experiments must be conducted before the conclusive statement can be made. For example, it is required to evaluate the mouse movement monitoring in different times, different grouping, and different environments/lighting.

Finally, I shall comment on the supplementary documents. The technical description in the supplementary files is confusing with poor reasoning. The figures and equations shown in the document are not clearly illustrated.

Reviewer #3:

Remarks to the Author:

I the manuscript "A Hierarchical 3D-motion Learning Framework for Animal Spontaneous Behavior Mapping" Huang et al., present a novel framework to study mouse behaviour by 3D visualization with multiple cameras. The authors used a combination of computational approaches to decompose small kinematics, learn about the dynamics and then provide a metric for mapping behaviours according to the features extracted.

The work is very well constructed, implemented and described in the manuscript. I have no doubt that it represents an advance in the field. It is outside my background a full understanding of specific details of this work. However, I have appreciated the potential and some of the limits. Here below a few comments that I hope will help to improve this work.

One of the problems we have in extracting behavioural features from visual-based systems is the quality of the picture, the contrast of lights and the occlusions. This is particular relevant when multiple animals are present in the same cage. These issues are not addressed in this work, and all 4 cameras acquire good quality images. I wonder whether the authors have considered to test the limit of their approach reducing the number of cameras (from the analyses).

The spatial structure of Figure 1c can be very useful in understanding mouse behaviour. I wonder to which extent this spatial structure is similar, or variable, across mice with the same background, whether the same mouse across temporal distant recording reproduces the same pattern and whether one can imagine patterns that are periodic, for example during the 24 hours.

The authors stated very clearly in different points of the manuscript that they based their study on a conceptual framework, which is that "behaviour adheres to a bottom-up hierarchical architecture". Regardless of some convenience, for example they report a two-stage decomposition, and in there one can appreciate some computational efficiency advantage, for example associated to redundancy in behaviours. However, I haven't understood how a bottom-up approach in this sense should provide a best match with neuronal codes. Are the authors suggesting that it will be possible, next step, to link fast neuronal activities to this temporal distinct behavioural architecture? If so, it seems to go against a parallel representation of the behaviour within the brain, do the authors want to comment on that?

Fig 4 and 6, the map of the behavioural phenotypes. How are the "fractions" defined? The authors seem to present a very detailed metric for dissecting behavioural features from, for example, different genotypes. I wonder how much this is a group effect and whether the same feature is present in all individuals of the same group? Are there any other combinatorial behaviours that present different clusters within the same group?

I believe that one of the strengths of this work is the decomposition of behaviour, which can then be tracked in time and used to predict modules of behaviour. I think that should this be applied extensively to behavioural studies will provide more convincing information about the validity. At the current state a longer monitoring in time, across days of individual animals would have provided, perhaps, a stronger validation of this framework.

One more last thing, although it is outside of the scope of this study, are the authors planning to extend this framework to mouse social interaction? As I said, this won't change what they have nicely achieved in this study, it is just a curiosity and an interest that involve everyone in the community.

Responses to the Review Comments (NCOMMS-20-43913)

We wish to thank two reviewers for their thoughtful and detailed reviews of our previous submission. These inputs prompted us to undertake the revision of our manuscript. This document provides a point-by-point response to the comments raised by the review panel. We believe the revised paper is better positioned, more focused, and makes a stronger contribution to the literature. We sincerely hope that you will find that this revised manuscript has improved substantially and is heading in the right direction. As the team shares several common comments, you may find some of our response repeated in the responses document. We feel that this approach makes it easier for each reviewer to read our response to his or her comment directly without jumping back and forth to different parts of our response document. Except for the language improvements, all other changes made to the **Manuscript**, **Supplementary Methods** and **Figure Legends** are highlighted in yellow.

Table of Contents

Reviewer #1 (R#1)	3
Reviewer #3 (R#3)	26
References	40

Reviewer #1 (R#1)		
COMMENT No.	REVIEW COMMENTS	AUTHORS' RESPONSES
R#1 (1)	This paper presents a parallel, multi-layered framework to learn the hierarchical dynamics and generate an objective metric to map the behaviour of a mouse into the feature space. Furthermore, 3D kinematics with the low-cost multi-view motion capture system have been analysed and discussed in this paper. The authors also demonstrated that the proposed technology can identify the animal behaviours and transgenic animal disease models from the behaviour monitoring. The major weakness of this paper is the lack of novel contribution. The proposed system mainly consists of a number of standard components developed by other researchers in the community. In addition, the bio-markers identified in this research lack convincing supportive evidence. The reviewer will further comment on these aspects in the following discussion.	We thank the reviewer for the summary of our work and also very much appreciate the reviewer's comments on the manuscript. Based on these suggestions, we have added new experiments, new data, and further analysis and validation. We address the reviewer's concerns point-by-point in the following text. We hope that these responses satisfy the reviewer's concerns. The revised manuscript has been improved based on the reviewer's suggestions. We have better presented the novelty of 3D motion capture system.  1) Highlighted our novelties by presenting the easy-to-use and robust solution for 3D animal behavior capture device. We extended the description of the fast calibration, synchronized multi-view video acquisition hardware. (related to Supplementary Fig. 2); 2) Proved that our system is a flexible and low-cost solution through the systematic validations for 3D animal behavior collection. These validations include: tested the limit of the system by reducing the number of cameras (related to Supplementary Fig. 4); validated the performance of 3D motion capture system in cases of body-part occlusion and viewpoint disappearance (related to Supplementary Fig. 5-6);
R#1 (2)	1. Novelty of the research:	Thank you for the insightful comments. We agree that such a description in the abstract may cause confusion. This sentence was intended to reflect one of the major contributions of our work: we are

The authors stated that "most recent end-to-end machine learning based behaviour analysis methods focused on recognizing behavioural identities in a static way or based on limited observations". The statement was based on limited survey on the related topics. In fact, a number of research projects have addressed dynamical and continuous activities of mice, e.g. (1)https://www.sciencedirect.com/science/article/pii/S0165027017301139, (2) (2)https://journals.plos.org/plosone/article?id=10.1371/journal.pone.0220751.	the first to propose a learning framework for motion dynamics inspired by the structure of natural behavior. Issues we address are the high-dimension of postural features, the large variability in the temporal scale, and the irregularity in the periodicity of the behavior primitives (movements) that increase the complexity of mammalian behavior. Therefore, we should first study the dynamic and hierarchical characteristics to decompose and discover the behavioral motifs, instead of just recognizing a given behavior feature sequence or using a static, fixed time window to segment behavior. According to your suggestion, we have rewritten a sentence to make this clear: - In Manuscript page 1, lines 19-21 (Abstract) However, the recent end-to-end machine-learning-based methods for behavior analysis mostly focus on recognizing behavioral identities on a static timescale or based on limited observations. We have carefully studied the two papers you referred. The continuous and long-term behavior recordings conducted in these papers has inspired us to conduct a continuous 24-hour recording experiment. The preliminary data reflects the circadian rhythm, proving that our framework has the potential for long-term monitoring and analysis. Moreover, our framework uses 3D multiple body-part posture tracking, which provides a more comprehensive and accurate representation of behavior and goes beyond the techniques used in these papers. The dynamic sub-second behavior decomposition of our method also has a higher time resolution for behavior identification than do the 20-sec moving windows used in one of the papers.
--	--

		As suggested, we added these papers as references and added the new of 24-hour recording data: - In Manuscript page 10, lines 322-327 (Discussion) For example, to study animal circadian rhythms, previous researchers have used electrophysiological and behavioral recording approaches to characterize different brain states^{1,2}. We used our framework to perform a continuous 24-hour behavioral recording, and the preliminary analysis proved that our framework could provide more comprehensive behavioral parameters and detailed quantification of behavior states (Supplementary Fig. 13).- In Supplementary Methods page 16, lines 337-366 (Supplementary Fig. 13 and figure legend) Supplementary Fig. 13 Continuous long-term monitoring and analysis of mouse behavior. a The timeline of the behavioral recording period over 24 hours. b The normalized velocity of the mouse across 24 hours aligned to the timeline. c The decomposed behavioral modules shown with color-coded labels. d Three magnified representative behavioral modules and selected, single corresponding frames. Left, running on the litter; Middle, eating; Right, prolonged immobility resembling resting. e, f State transitions of the movement modules in night and day phases. g Differences in the state transitions between night and day. The color of the dots in e, f, and g correspond to the behavioral modules shown in c. The size of the dots represents the rank of the module probabilities over 24 hours. The color of the connections in e and f represents the direction from the previous state to the current state, and its color is the same as that of the previous state. The width of the connections in e and f
--	--	--

		represents the normalized two-state transition probability. The color and width of the connections in g represent the normalized difference between e and f.
R#1 (3)	(1.1) 3D motion capture: The 3D motion capture system is based on the standard apparatus, camera calibration, pose estimation (Ref 17), and 3D skeletal reconstruction (Ref 41). It is reasonable to include the state of art technologies at this stage. However, the issues are, the stages of pose estimation/3D skeletal reconstruction miss the investigation on body occlusion and view-point disappearance.	Thank you for the suggestion. In our framework, accurate behavioral data acquisition is necessary. Recently, many new approaches have emerged in animal behavioral tracking. We appreciate that the existing excellent techniques and tools provide us with the opportunity to attempt a dynamic and hierarchical behavior decomposition as implemented in our framework. Although these technologies of pose estimation and 3D reconstruction have the potential for obtaining 3D postural time-series, there has been until now a lack of practical and accessible software and hardware solutions for the issues of fast calibration, synchronized video acquisition, and accuracy verification. During the development of our framework, we invested much effort in identifying, verifying, and improving various discrete hardware and software modules. We hope to provide experimenters with a complete, easy-to-use, and robust solution to make 3D behavior collection less labor-intensive. Therefore, we extended the description of the novelty and validation of the 3D motion capture system as follows:  - In Manuscript page 4, lines 111-119 (Results) To efficiently and comprehensively characterize the kinematics of free-moving animals, we developed a 3D multi-view motion capture system (Fig. 2a, b) based on recent advances in techniques for pose estimation and 3D skeletal reconstruction. The most critical issues in 3D animal motion capture are efficient camera calibration, body

		occlusion, and viewpoint disappearance, which have not been optimized or verified. To address these issues, we developed a multi-view video capture device (Supplementary Fig. 2a). This device integrates the behavioral apparatus, an auto-calibration module (Supplementary Fig. 2b, d), and synchronous acquisition of multi-view video streams (Supplementary Fig. 2c). While the conventional manual method requires half an hour to produce the required checkerboard for calibration, the auto-calibration module can be completed in one minute. - In Supplementary Methods page 3, lines 51-73 (Supplementary Fig. 2 and figure legend) Supplementary Fig. 2 Illustration of the multi-view video capture device and the workflow of the auto-calibration module. a Schematic of the multi-view video capture device. The support framework is a $90 \times 90 \times 75 \text{ cm}^3$ movable stainless steel shelf, on which cameras, behavioral apparatus, calibration modules, and background lighting are mounted. A shielding curtain can be added per experimental requirements. b The auto-calibration module is designed for efficient camera calibration in 3D and is composed of an LCD screen for displaying the checkerboard and a control unit used to tilt the screen. To collect images of the checkerboard pattern at different orientations relative to the cameras, the calibration program controls the screen to rotate and translate the checkerboard pattern at different tilt angles. With this auto-calibration module, the checkerboard images can be captured in one minute. c The multi-view video acquisition module. Four video streams, one per camera, are input to the PCI-E USB-3.0 data acquisition card (expanded bandwidth). The acquisition program then uses multi-thread
--	--	---

		acquisition to ensure frame synchronization. d The two-part workflow of the auto-calibration program. The first part, shown on the left, automatically collects a variety of checkerboard patterns for each camera (70). Right, the calibration process, which is based on the MATLAB StereoCameraCalibrator GUI. Regarding the critical issues of body occlusion and viewpoint disappearance that you mention, our approach avoids these issues in three ways: 1) When constructing the DLC training set and when body parts were invisible, we guessed and labeled their location in each single view. Therefore, the model can predict invisible body parts and output a confidence score as a likelihood to help us identify whether the predicted body parts in the current frame are reliable;2) Before 3D reconstruction, for each body part, we specify a likelihood threshold to help determine how many cameras are needed to reliably obtain the 2D coordinates of the body part;3) In theory, 3D reconstruction can be achieved as long as any two cameras can obtain the 2D coordinates of the same point in 3D space from different views. Since we acquire the animal's behavior images with four complementary views, this guarantees that any body part can be captured by at least two cameras with high probability. To demonstrate that our system can reliably track occluded and viewpoint-disappeared 3D body parts, we have added data and extended the verification of these two points as follows: - In Manuscript page 4-5, lines 126-140 (Results)
--	--	---

		We next investigated whether the 3D motion capture system could reliably track the animal in cases of body-part occlusion and viewpoint disappearance. We checked the DeepLabCut (DLC) tracking likelihood in the collated videos (0.9807 ± 0.1224, Supplementary Fig. 4a) and evaluated the error between the estimated 2D body parts of every training set frame and the ground truth ($0.534 \pm 0.005\%$, Supplementary Fig. 5b). These results indicated that in most cases, four cameras were available for 2D pose tracking. Since 3D reconstruction can be achieved as long as any two cameras obtain the 2D coordinates of the same point in 3D space from different views, the reconstruction failure rate caused by body occlusion and viewpoint disappearances is determined by the number of available cameras. Therefore, we evaluated the average proportion of available cameras in situations of body part occlusion and viewpoint disappearance. The validation results for body-part occlusion show an average reconstruction failure rate of only 0.042% due to body occlusion or inaccurate body-part estimation (Supplementary Fig. 5c). While for viewpoint disappearances, both tests (Supplementary Fig. 6, and Supplementary Video 4, 5) proved that our system has a high reconstruction rate for animal body parts. Moreover, the artifact detection and correction features can recover the body parts that failed to be reconstructed. Body occlusion: - In Supplementary Methods page 6, lines 136-156 (Supplementary Fig. 5 and figure legend) Supplementary Fig. 5 3D reconstruction process and reliability evaluation of the occluded body parts. a The workflow of 3D reconstruction of a single body part: 1) estimate the two-dimensional
--	--	--

		coordinates of the animal's body part from four cameras; 2) select the cameras to be used for reconstruction by thresholding the likelihood of the estimated body part; 3) determine whether the number of cameras available meets the reconstruction requirements (2 or more); and 4) if 2 or more, reconstruct the 3D coordinate of the body part. Otherwise, the 3D reconstruction fails due to the occlusion. P1, primary camera. S1, first secondary camera. S2, second secondary camera. S3, third secondary camera. b The errors in 2D body-part estimations versus ground truth. The error rates are shown for each body part separately in the boxplot and averaged $0.534 \pm 0.005\%$. c The proportional number of available cameras by body part. The average proportions are: no cameras, 0.004%; 1 camera, 0.015%; 2 cameras, 0.038%; 3 cameras, 1.048%; 4 cameras, 98.895%. View-point disappearance: - In Supplementary Methods page 7-8, lines 159-193 (Supplementary Fig. 6, figure legend, and Supplementary Video 4, 5) Supplementary Fig. 6 Evaluation of the 3D reconstruction in cases of view-point-specific disappearances of body parts. a, d 2D pose tracking and 3D skeleton reconstruction of representative view-point disappearance frames from two different test apparatuses. a First test: square open field test. The behavior chamber is out of the field of view, and blind areas may occur when the animal enters the four corners; d Second test: circular open-field with a sociability cage. The mouse can easily be occluded by the cage, thus blind areas may exist in one or more perspectives. Top: selected frames with one or more views in which body parts disappear. Middle: magnification to show the disappearance details. Bottom: successfully reconstructed
--	--	---

		3D skeletons shown in approximately the same views as the corresponding recordings. b, e The proportional number of cameras available for 3D reconstruction for each body part. In the first test, an average of $99.398 \pm 0.149\%$ of all frames showing the body part can meet the reconstruction requirements; In the second test, the average reconstruction rate of all body parts is $99.776 \pm 0.150\%$. c, f The distribution of the number of available cameras for 3D reconstruction. The color-coded dots indicate how many cameras are used for reconstruction at the indicated location. The positions of all the points are the x and y coordinates of the nose. For visualization purposes, the data are down-sampled to 10%. The proportional number of cameras used to reconstruct the nose in 3D is indicated in the key on the left.
R#1 (4)	(1.2) Non-locomotor movements with dynamic time alignment Kernel: Center alignment and rotation transformation have been applied in the pre-processing stage. This process is tedious indeed. Afterwards, a temporal reduction algorithm is used to merge the adjacent similar poses, which group wrong poses into individual pose categories. The standard DTAK method is also used to measure the similarity between sequences. However, it is not clear why not use the standard DTW method in this case.	Thank you for the comments. We agree that the center alignment and rotation transformation are tedious and are just necessary pre-processing. Temporal reduction in poses: Regarding pose decomposition: First, most previous reviews ^{3,4} and studies ⁵ believe that the basic element of animal behavior is the pose. The pose is a static snapshot of a movement sequence at any moment, and the movements are stereotyped spatial-temporal patterns encoded by a certain number of posture sequences. In nature, animal movement is continuous, and due to the high dimensionality of the mammalian skeleton, the number of behavioral posture variables is potentially infinite ⁶. On the other hand, adjacent poses are usually highly correlated and

		redundant for purposes of behavior quantification and analysis ⁷. This is particularly evident in the 24-hour recording: when the mouse was resting, it maintained a constant posture for a long time. In such cases, the overall behavioral state can be represented by a single pose and its duration. Second, the behavior decomposition part of our framework is required to search for and align potential movement clusters dynamically. For computational efficiency (the computational complexity of behavioral decomposition is $O(n^2)$), the algorithm allocates more computing resources to rapidly changing, dynamic information than it does to static information. This temporal reduction greatly improved the efficiency of our supplementary analysis of the 24-hour long-term recording (Supplementary Fig. 13). Third, although the movement decomposition is based on temporally reduced postural sequences, the clustering quality evaluation (Fig. 4D) is based on the original postural time-series generated segment kernel matrices. The result reflects a higher intra-cluster homogeneity and the inter-cluster heterogeneity. We modified the paper as follows: - In Manuscript page 5, lines 149-152 (Results) Animal movement is continuous, and due to the high dimensionality of the mammalian skeleton, the behaviorally relevant posture variables are potentially infinite in number. However, adjacent poses are usually highly correlated and redundant for behavior quantification and analysis, which is particularly evident in long-term recording.
--	--	--

		Why not use the standard DTW? Thank you for the question. An appropriate distance metric is critical for achieving optimal segmentation of a continuous postural time-varying series. Initially, we used DTW directly because it has been successfully applied to aligning series data such as speech, ECG, and gene sequences ⁸. However, as pointed out in the literature, DTW does not satisfy the triangle inequality, limiting its application to that of a distance metric for calculating similarity between samples. The DTAK proposed later satisfies the Cauchy-Schwartz inequality and is used as the kernel of the SVM to recognize speech ⁹. Hence, we adopted DTAK into our framework for measuring the similarity between the behavioral segments. We added and cited references to explain this point:  - In Manuscript page 5, lines 163-168 (Results) An appropriate distance metric is critical for modeling the temporal variability and optimizing the NM segmentation of a continuous postural time-varying series. Although dynamic time warping (DTW) has commonly been applied in aligning time-series data, it does not satisfy the triangle inequality. Thus, we used the improved DTAK method to measure the similarity between time sequences and construct an energy equation (objective function) for optimization.
R#1 (5)	(1.3) Mapping mouse movements with low-dim embeddings and unsupervised clustering: A standard UMAP method is used to preserve both the local and global structure of the dataset. However, it is not clear to the reader why this	Thank you again for the reminder. Evidence to support UMAP preservation of both local and global structure:

	method must be used - lacks supporting evidence. BIC is used to model the structure but how?	Following the two-stage behavior decomposition, we use DTAK to construct the structural representation (n-by-n distance matrix) of relationships among the movement sequences, a high-dimensional space in which humans have difficulty imagining and understanding the similarities or differences. Therefore, performing dimensionality reduction on the distance matrix can provide an informative visualization of this structure in a low-dimensional (low-D) space. We evaluate the results of dimensionality reduction by 2 criteria ^{10,11}: 1) It should be able to represent the local structure to facilitate subsequent clustering, which means similar movement sequence can be grouped together as neighbors;2) It should be able to preserve the global structure, which can be reflected in the correlation between the low-D representation and the original distance matrix. Generally, the low-D representation should preserve the intra-cluster homogeneity and the inter-cluster heterogeneity after dimensionality reduction. To this end, we added the results of evaluation of the three commonly used dimensionality reduction algorithms (UMAP, PCA, and tSNE) from these two aspects. - In Manuscript page 6, lines 197-206 (Results) However, since the 936-D matrix cannot provide an informative visualization of behavioral structure, it is necessary to perform DR on this data. Various DR algorithms have been designed either to preserve the global representation of the original data or to focus on local neighborhoods for recognition or clustering. Thus, in animal behavior quantification, we face a trade-off between discretizing behavior to provide a more quantitative analysis, and maintaining a global representation of behavior to characterize the neural-behavioral
--	---	--

		relationship's potential manifolds. Therefore, we first evaluated the commonly used DR algorithms from the standpoints of preserving either the global or the local structure. The evaluation results show that UMAP can balance both aspects for our data (Supplementary Fig. 8) and provide 2D embeddings of these NM segments. - In Supplementary Methods page 10, lines 218-244 (Supplementary Fig. 8 and figure legend) Supplementary Fig. 8 Comparison of three algorithms of dimensionality reduction for the representation of the NM feature space structure. a Segment kernel matrix of a representative single-session behavioral experiment recording. The matrix pixels represent the normalized similarity value of 937 pairs of decomposed movement segments. b-d Dimensionality reduction with the three most-used algorithms: UMAP, tSNE (t-distributed stochastic neighbor embedding), and PCA (principal component analysis). For visualization purposes, the segment kernel matrix is reduced to two dimensions. e Quantification of local structure preservation by evaluation of the silhouette criterion values of the dimensionality reduction result of each algorithm. The silhouette criterion values are calculated by enumerating the clusters from two to twenty. The average silhouette criterion values are: UMAP, 0.557 ± 0.011; tSNE, 0.619 ± 0.021; PCA, 0.240 ± 0.039. f Quantification of global structure preservation by evaluation of the Spearman correlation coefficients between the original segment kernel matrix and the dimensionality-reduced result of each algorithm. For each algorithm, we first randomly subsampled 70% of the kernel matrix 20 times. Each time, the Spearman correlation coefficients are calculated between the selected segment kernel sub-matrix and the paired-wise
--	--	---

		distances of the dimensionality-reduced data. The average coefficients are: UMAP, 0.817 ± 0.001; tSNE, 0.326 ± 0.001; PCA, 0.913 ± 0.002. ****, $P < 0001$ by two-way ANOVA with a Holm-Sidak post-hoc test. Response to comment re BIC: We are sorry for the confusion. In our manuscript, use of the BIC was intended to determine the number of clusters into which the decomposed movement sequences should be partitioned. Our framework adopts an unsupervised strategy, and we most unsupervised clustering requires a pre-specified cluster number. This issue has been highlighted in many behavioral studies and reviews ¹². The solution to this problem can be data-driven ¹³ or refer to the context of the practical biological problem. Here, we chose the data-driven approach. We assumed that the constructed behavior feature space consists of a finite number of Gaussian mixture states, and our task was to estimate the optimal number of mixture states. Specifically, we adopted the clustering analysis function mclust from the R package ¹⁴, and we used 14 models with it for our data estimation. The BIC of each model is calculated based on a given number of states, and is then obtained for all models for all numbers of states. Finally, we chose as the optimal number of clusters the number of states that allows the largest number of models to achieve the largest BIC. We have rewritten the sentence to make this clear: - In Manuscript page 7, lines 209-214 (Results) We used an unsupervised clustering algorithm to investigate the behavior's spatio-temporal representation and identify the movement
--	--	--

		phenotypes. Most unsupervised clustering require a pre-specified number of clusters, and the number chosen can be data-driven or refer to the context of the practical biological problem. In the single experimental data shown in Figure 4a, the data-driven Bayesian Information Criterion in the R package mclust was adopted to determine that the optimal cluster number was 11 (Supplementary Fig. 10).
R#1 (6)	(1.4) Kinematic validation of mouse behavioural phenotypes: MI components have been constructed to describe the postural difference but 'why and how' are not explained.	Thank you for the comment. To verify whether our framework can categorize behaviors accurately, we provided further validation from the kinematics aspect, in addition to the intra/inter-cluster correlation coefficient evaluations (Fig. 4 c-e) based on the segment kernel matrix. These further validations include:  1) Manual inspection of the segmented video clips; 2) Visualizing the average-skeleton (common poses) of each category; 3) Quantifying the main kinematic parameters. In Fig. 5a, we first visualized the average skeleton and the overlaid heatmap of the distribution and movement intensity (MI), showing homogeneous patterns of the identified behavioral phenotypes. Then, we used MI as a metric to compare the kinematic characteristics of each body part in each behavioral phenotype. MI is related to velocity, and contains both horizontal and vertical components. Generally, locomotion (e.g., walking, trotting, and running) has a larger horizontal MI, while rinsing, rearing, and stretching have higher vertical MIs. Moreover, the MI can also reflect the fact that that some body parts have greater freedom of motion, such as the nose and front paws (Fig.). These results indicate that MI can

		effectively represent the same visual information as in the validations of steps 1) and 2). We realized that the rationale and calculation process of MI should be clearly described, so we added an explanation and description of this part: - In Manuscript page 7, lines 236-239 (Results) To provide a kinematic validation of the identified behavioral phenotypes from the perspectives of visualization and quantification, we first visualized the average skeleton, which was averaged over all frames in each movement cluster (Fig. 5a).- In Manuscript page 7-8, lines 241-245 (Results) The detailed kinematic parameters, especially the velocity of each body part, could provide greater sensitive differences than the unclear visually-based assessments. Therefore, we defined movement intensity (MI) as a metric for characterizing the kinematics of each body part in each behavioral phenotype (see Supplementary Methods for further details). MI is related to velocity, and it contains both horizontal and vertical components.- In Manuscript page 23, lines 744-749 (Fig. 5 figure legend) c The comparison of MI between different movement phenotypes. Each movement segment has two MI components (red boxes, horizontal; blue boxes, vertical). The boxes' values for each group contain the MIs of n behavioral modules (n is the number of behavioral modules of each group). d, e Horizontal and vertical MI of each body part in different movement phenotypes. The values on each line are the MIs of all behavior modules corresponding to the
--	--	--

		phenotype, shown by body parts separately and presented as mean \pm standard deviation (SD).
R#1 (7)	2. Identification of behavioural signatures of mouse disease models: Two groups of mice have been used to investigate possible bio-markers. 6 KO and 6 WT mice may suggest something but more substantial experiments must be conducted before the conclusive statement can be made. For example, it is required to evaluate the mouse movement monitoring in different times, different grouping, and different environments/lighting.	Thank you for the comment. According to your suggestion, we have added 72 behavioral data in total, which include: new data on 5-6 week old, male mice in white light in the circular open-field test, thus 4 samples in each group (KO and WT); We added new groups according to different experimental apparatus, lighting conditions, ages, and sexes, 8 KO and 8 WT mice for each condition. Therefore, the behavioral dataset has been updated with ten groups of 84 cases (Supplementary Table 2). Updated data on 5-6 week, male mice in white light, circular open field We updated the analysis to account for the new data, with the result that:  1) The hunching behavior of the KO mice was still significantly higher than that of WT, and the comparison of the kinematic parameters of hunching and rearing was also consistent with the previous data (Fig. 6g, and j-n). 2) Three subtypes (M39, M40, and M41 in Fig. 6g) of self-grooming behavior were identified. We manually checked the videos corresponding to these subtypes and found their poses had slight differences (e.g., head orientations are different). Although our unsupervised framework unbiasedly categorized them into distinct behavior modules, they are well-sorted on the dendrogram according to their similarity (Fig. 6g), which helped us to merge and annotate them easily.

		3) With greater numbers of samples, behavioral differences that were not previously significant became so. We found that the Shank3B KO mice showed an increase engagement in hunching and self-grooming stereotyped behaviors, accompanied by deficiencies in other behaviors, stepping (M5), walking (M14), and two (M21 and M22) subtypes of rising behaviors. This result indicates that locomotion and vertical movement in KO mice are less than in WT mice (consistent with the average velocity comparison shown in Fig. 6b). - In Manuscript page 8, lines 272-275 (Results) By manually reviewing the video clips of these four types, we annotated the 38th movement (M38 in Fig. 6g) as hunching; we also found that three of the movements were very similar (closely arranged on the behavioral dendrogram, Fig. 6g). Therefore, we grouped them and annotated them as self-grooming. - In Manuscript page 9, lines 285-289 (Results) Besides the four phenotypes that KO mice preferred more than the WT mice did, the KO mice also showed four additional deficit behavioral phenotypes, namely stepping (M5), walking (M14), and two types of rising (M21 and M22). This result indicates that the locomotion intensity and vertical movement of KO mice were lower than those of WT mice. Group comparison of Shank3B mice under different conditions: We performed 3D reconstruction, behavioral decomposition, and re-clustering on all collected samples to convert the continuous
--	--	---

		behavioral recording data of each sample into a fractional prevalence of each behavioral phenotype. These 84 samples constitute a movement fraction matrix. Due to the large number of added conditions and samples, we performed a preliminary analysis based on the movement fraction matrix at the group level. By calculating the cross-correlation coefficients and extracting the principal components (Supplementary Fig. 12), we found that: 1) For male mice of the same genotype, changing only a single experimental condition caused no significant differences in group behavior patterns; Although there was no significant difference in group behavior patterns between male and female Shank3B KO mice under the same experimental conditions, females showed a weak autism-like behavioral tendency (Supplementary Fig. 12d).2) With all other conditions the same, we found that when comparing the behavioral patterns of the different genotypes KO and WT, only the female group showed no significant difference, while significant differences were found between KO and WT mice under all other conditions. We showed that these results are consistent with previous reports¹⁵ that Shank3B KO male mice display more severe impairments in motor coordination than do females. - In Manuscript page 10-11, lines 340-356 (Discussion) Moreover, we further investigated the differences in the behavior patterns of Shank3B KO and WT mice at the group level. In addition to the data that had already been analyzed (collected under the condition: male mice, 5–6 weeks, white light, and circular open-field), we extended the group behavioral pattern analysis to include
--	--	--

		data collected under different conditions (i.e., different experimental apparatus, lighting, age, and gender; Supplementary Table 2). We calculated the cross-correlation coefficient matrix (CCCM) of all samples based on the movement fractions and used principal component analysis to extract the main variance factors of the CCCM (Supplementary Fig. 12 a, b). We found that when only a single condition was changed for male mice, there was no significant difference in population behavior patterns in mice with the same genotype (Supplementary Fig. 12 c). We also found that although some female KO mice had a weak tendency for autistic-like behavior, there was no significant difference between 5–6 week male and female KO mice at the group level when tested under the white-light circular open field condition (Supplementary Fig. 12 c, d). Finally, we compared the behavior patterns when all conditions were the same except for the genotypes. The results showed that only the female group showed no significant difference between KO and WT genotypes, while significant differences in behavioral patterns were found between KO and WT mice under all other conditions. These findings are consistent with previous reports that Shank3B KO male mice display more severe impairments than females do in motor coordination. - In Supplementary Methods page 14-15, lines 302-335 (Supplementary Fig. 12 and figure legend) Supplementary Fig. 12 Group comparison of Shank3B KO mice under different conditions. a The movement fraction matrix of mice in ten different groups. The color-bars shown in the left indicate the group conditions for the corresponding rows of the matrix (see Supplementary Methods for further details). Each row in the fraction
--	--	---

		matrix represents the tested mouse, and each column corresponds to the behavioral module types arranged with the dendrogram in the bottom. For visualization and comparison purpose, the values of movement fraction matrix are normalized with z-score by rows. In each group, the row orders are determined by placing the sample with largest variance of the movement fraction, and then the other samples are ranked according to the decreasing correlation with the first row. b The cross-correlation coefficients matrix (CCCM) of the movement fractions among all ten groups samples. c The group comparisons of behavioral correlations between the selected conditions, which are shown with twelve submatrices of b. d The behavioral statistics between ten groups. The comparison metric is determined by calculating the principal component (PC) of the CCCM, then using the first PC (PC1) to evaluate the overall behavioral differences across ten groups (Kruskal-Wallis test, **, $p < 0.01$, ***, $p < 0.001$).
R#1 (8)	Finally, I shall comment on the supplementary documents. The technical description in the supplementary files is confusing with poor reasoning. The figures and equations shown in the document are not clearly illustrated.	We have followed your suggestions and made many improvements to the supplementary materials for clarity. These improvements include:  1) Adding a new figure to illustrate our framework's general workflow (Supplementary Fig. 1) and describing the key steps and their corresponding descriptions in the remainder of the Supplementary Methods in the Figure Legend. Through this figure, we tried to clearly show the roles in our framework and the inputs/outputs of the steps described in Supplementary Methods. 2) Merging the figures in the preprocessing section to make it more concise and clear. 3) Emphasizing the purposes of those methods to better show the reasoning. We have highlighted these modifications in the supplementary files.

		- In Supplementary Methods page 1-2, lines 1-48 (Supplementary Fig. 1) Supplementary Fig. 1 The workflow of the hierarchical 3D-motion learning framework. a Four main steps for a single experimental session: 1) Calibration (related to Supplementary Methods “The calibration of 3D motion capture system”). Using the auto-calibration module to quickly prepare 70 groups of checkerboard images from various angles and positions for calibration and using the MATLAB StereoCameraCalibrator GUI to calculate the calibration parameters of the three pairs of cameras. This step is necessary only when the calibration parameters are unknown, or the cameras have been moved. 2) Data collection (related to Supplementary Methods Animals, behavioral experiments and behavioral data collection). Setting up the behavioral apparatus and preparing the animal and then using the multi-view video capture device to collect the synchronous behavioral videos. 3) 3D reconstruction (related to Supplementary Methods 3D pose reconstruction). Using the DLC pre-trained model to predict the animal’s 16 body-part 2D coordinates from the four separate videos, then performing the 3D skeleton reconstruction with the 2D coordinates from four views to obtain the animal’s postural time-series. 4) Behavior decomposition (related to Supplementary Methods Behavior decomposition). Performing the two-stage behavior decomposition on the pre-processed postural time-series. This step discovers the behavioral modules based on the optimal movement segmentation. Finally, these behavioral segments are aligned using the DTAK metric to construct the segment kernel matrix representing their similarity. b Group analysis based on specific biological questions. 1) Merge and dimensionality reduction
--	--	--

		(related to Supplementary Methods: Group segment kernel matrix and low dimensional embedding). According to experimental grouping, single session segment kernel matrices are merged into a group segment kernel matrix. To visualize the informative structure of the behavioral modules involved, we used dimensionality reduction to transform the group segment kernel matrix into a 2D space. 2) Unsupervised clustering (related to Supplementary Methods: Unsupervised clustering). Constructing the behavioral map by combining the NM space with the locomotion dimension, then using the unsupervised clustering algorithm to categorize the movement sequence into distinct types. After clustering, ethograms can be constructed by associating the behavioral labels with their original segments. 3) Downstream analysis. After obtaining each session's ethogram, the downstream quantitative analysis can be conducted according to experimental grouping, recording stage, and other conditions to answer biological questions from behavioral aspects.
--	--	---

Reviewer #3 (R#3)		
R#3 (1)	I the manuscript “A Hierarchical 3D-motion Learning Framework for Animal Spontaneous Behavior Mapping” Huang et al., present a novel framework to study mouse behaviour by 3D visualization with multiple cameras. The authors used a combination of computational approaches to decompose small kinematics, learn about the dynamics and then provide a metric for mapping behaviours according to the features extracted. The work is very well constructed, implemented and described in the manuscript. I have no doubt that it represents an advance in the field. It is outside my background a full understanding of specific details of this work. However, I have appreciated the potential and some of the limits. Here below a few comments that I hope will help to improve this work.	Thank you for your appreciation and valuable suggestions for our manuscript, which were very helpful in improving our framework to promote the understanding of behavior. We have added experiments, further analysis, and further validations to address the points you raise. We trust that these responses fully satisfy the reviewer's concerns. The manuscript has been improved because of the reviewer's suggestions.
R#3 (2)	One of the problems we have in extracting behavioural features from visual-based systems is the quality of the picture, the contract of lights and the occlusions. This is particular relevant when multiple animals are present in the same cage. These issues are not addressed in this work, and all 4 cameras acquire good quality images. I wonder whether the authors have considered to test the limit	Thank you for the comment. The main purpose of the multi-view behavior capture system is to collect high-quality behavior pictures, as a single fixed view may easily be limited by animal body occlusion and blind areas. Regarding the contrast of lights, due to the strong adaptability of the pose estimation toolbox (DLC) we used, animal tracking can be achieved only by labeling training set under different contrast conditions. For occlusions, the advantage of multiple views is to provide complementary information, thereby improving the

	of their approach reducing the number of cameras (from the analyses).	robustness of animal pose tracking. As shown in Supplementary Fig. 5a, 3D reconstruction of a single body part requires at least two cameras with different views to simultaneously obtain its 2D coordinates. However, for multiple body-part 3D reconstruction, or in the complex experimental scenes shown in Supplementary Fig. 6, we find that some views of the four cameras have blind areas or the body is occluded by objects in the arena. In these cases, there are only two views available for 3D reconstruction, and in some locations, only one camera can achieve reliable tracking, causing failure of 3D reconstruction. Therefore, in actual experiments, the number of cameras is a tradeoff between the complexity of the experimental environment and the hardware accessibility. According to your suggestion, we have further verified the influence of reducing the number of cameras on the 3D reconstruction. - In Manuscript page 9-10, lines 312-316 (Discussion) The multi-view motion capture system can avoid animal body occlusion and view-angle bias and estimate the pose optimally by flexibly selecting the view to use according to the tracking reliabilities of the different views. We also confirmed the necessity of using multi-view cameras in complex experimental scenes, whereas in the simple experimental scenes, only three or even two cameras were needed (Supplementary Fig. 4).- In Supplementary Methods page 5, lines 106-133 (Supplementary Fig. 4 and figure legend) Supplementary Fig. 4 Evaluation of 3D Reconstruction Quality with Different Camera Settings. a The likelihoods of the DLC pose
--	--	--

		estimations of four camera positions. P1, primary camera 1, S1, secondary camera 1, S2, secondary camera 2, S3, secondary camera 3. Each point on the boxplot represents the mean likelihood of each test recording, which is calculated by firstly averaging the likelihoods of all the body parts per frame then averaging them across all frames. The likelihoods show no significant differences among these cameras (Kruskal-Wallis test, $p = 0.1339$, $n = 16$). b The likelihoods of the 3D reconstructions of different camera groupings. 2C180, two cameras are placed in opposite directions. 2C90, two cameras are positioned in orthogonal directions. 3C, three cameras. 4C, four cameras. In the camera groupings 2C180 and 2C90, each point on the boxplot is calculated by firstly specifically averaging the likelihoods of two paired body parts for calibration, then averaging all 16 paired averaged likelihoods per frame and finally averaging them across all frames. In the camera groupings of 3C and 4C, each point on the boxplot is calculated by firstly specifically averaging the first two maximum likelihoods of paired body parts for calibration from all the three or four points, then averaging all 16 paired averaged maximum likelihoods per frame, and finally averaging them across all frames. c The variances of the behavioral trajectories captured by different camera groupings. Each point on the plot is calculated by firstly computing the variances of each body part's trajectory in the X, Y, Z axis, then averaging them across X, Y, Z axis, and finally averaging them across 16 body parts. (Kruskal-Wallis test, $****p < 0.0001$, $n = 16$). d The variances of each body part in X, Y, Z coordinates of varying camera groupings. The variances of each body part are calculated by firstly computing the variances of each body part's trajectory in X, Y, Z axis then averaging them across X, Y, Z axis.
--	--	--

R#3 (3)	The authors stated very clearly in different points of the manuscript that they based their study on a conceptual framework, which is that “behaviour adheres to a bottom-up hierarchical architecture”. Regardless some convenience, for example they report a two-stage decomposition, and in there one can appreciate some computational efficiency advantage, for example associated to redundancy in behaviours. However, I haven’t understood how a bottom-up approach in this sense should provide a best match with neuronal codes. Are the authors suggesting that it will be possible, next step, to link fast neuronal activities to this temporal distinct behavioural architecture? If so, it seems to go against a parallel representation of the behaviour within the brain, do the authors want to comment on that?	Thank you for your insightful comment. One of our motivations for developing this framework was to better understand the relationship between neural activity and behavior, as you mention. We designed the behavior decomposition framework according to a bottom-up architecture since we were mainly considering the convenience of the concept description as well as the implementation and performance of the computation. Owing to the complexity of mammalian naturalistic behavior and the lack of well-annotated behavioral databases, our framework’s strategy is to automatically discover potentially meaningful spatio-temporal patterns (namely behavioral modules/primitives/motifs) from the recorded continuous postural time-series instead of detecting behaviors based on predefined rules. Therefore, the feasible solution is to process the low-level postural features first and then reveal the behavioral structure as a self-organized behavioral feature space. In terms of the relationship between behavior and neural activity, it remains an open question. According to our literature research, the neural-behavioral relationship can be interpreted from the two aspects of behavior control (specifically motor behavior) and neural representation. From the behavior controlling perspective, the hierarchical theory holds that neural activity controls behavior top-down. The higher-level units, such as the motor cortex, control lower-level motor units (e.g., spinal cord or brainstem) by initializing and sending motion commands to coordinate posture and balance to perform the movement ¹⁶. Under this theory, neural activity can generate parallel behavioral patterns (e.g., walking and chewing gum at the same time ⁴). While from the neural representation perspective,
---------	--	--

		behavior generation and representation may not share the same structure and neural activity^{17,18}. Moreover, the behavior may not be represented by a single neuron or independent neural activity but by a manifold that represents a few latent variables in the population. Therefore, parallel behaviors may be represented as the superimposed representation of each independent mode or as a new neural activity pattern. The constraints of animal behavior in the current research paradigm also limit the capture of neural-behavioral covariates¹⁹. Even if advanced recording technology allows us to record large population activities with higher temporal-spatial resolution, this may not lead to new insights. Thus, only by combining large population activities with the accurate measurement and identification of naturalistic, complex behavior can we unravel the essential rules. As you commented, we next step will first focus on the collected large sample Shank3B KO disease model, build a well-annotated behavior database to involve more researchers in this community. Then combining our framework with free-moving two-photon microscopy and electrophysiological recording links the neural activity patterns and functional connections with the cross-scale behavioral dynamics and timing patterns. However, I have to say that current behavioral quantification approaches, including our framework, are still evolving. There are still unsolved issues, such as defining and decomposing the co-occurring behavior and interpreting the neural-behavioral relationship with new algorithms. As suggested, we have added discussion on this point. - In Manuscript page 11, lines 367-375 (Discussion)
--	--	--

		In other words, to understand the encoding/decoding relationship rules of the neural activity generating behavior and behavior's neural representation, synchronization of large population activities and accurate measurement and identification of naturalistic, complex behavior are required. In the future, we will focus on combining our framework with free-moving two-photon microscopy and electrophysiological recording to link the neural activity patterns and functional brain connections with the cross-scale behavioral dynamics and timing patterns. Therefore, with further technical optimization and the open-source of a large sample, well-annotated disease model behavior database open source, our framework may contribute to resolving the relationships between complex neural circuitry and behavior, as well as to revealing the mechanisms of sensorimotor processing.
R#3 (4)	Fig 4 and 6, the map of the behavioural phenotypes. How is the “fractions” defined? The authors seem to present a very detailed metric for dissecting behavioural feature from, for example, different genotypes. I wonder how much this is a group effect and whether the same feature is present in all individuals of the same group? Are there any other combinatorial behaviours that present different clusters within the same group?	Thank you for the comment. How are the “fractions” defined? For each subject, the behavior fractions are defined as the bouts number of each behavioral phenotype divide by the total number of behavior bouts the animal produced during the experiment. For example, in Fig. 4c, the fraction of running is $26/935=0.0278$; In Fig. 6I, the grooming behavior of mouse KO-1 occurs 35 times, and the total number of behavioral modules of this mouse is 794. Therefore, the grooming fraction of mouse KO-1 is 0.0441. We added the definition of fractions and provided a statistical result based on the metric of a behavior modules' duration:  - In Manuscript page 22, lines 707-710 (Figure legend of Fig. 4)

		c Behavior fractions. For each subject, the behavior fractions are defined as the bout number of each behavioral phenotype divided by the total number of behavior bouts the animal produced during the experiment. - In Manuscript page 23, lines 737-749 (Figure legend of Fig. 5) c The comparison of MI between different movement phenotypes. Each movement segment has two MI components (red boxes, horizontal; blue boxes, vertical). The boxes' values for each group contain the MIs of n behavioral modules (n is the number of behavioral modules of each group). d, e Horizontal and vertical MI of each body part in different movement phenotypes. The values on each line are the MIs of all behavior modules corresponding to the phenotype, shown by body parts separately and presented as mean \pm standard deviation (SD). - In Supplementary Methods page 29, lines 738-748 (Supplementary Methods) After each MI of body parts in XY (horizontal) and YZ (vertical) coordinates plane were calculated, they were visualized in Fig. 5. We visualized them in specific top view (XY) and side view (YZ). Firstly, we averaged the positions of each body part across time to plot averaged pose skeleton of each movement category by solid line. Secondly, we meshed the plane of averaged pose skeleton to create grids for visualizing MI. Each position of skeleton has a corresponding MI parameter, which could be plotted on the grids. The MIs in the same grids are averaged to get the mean MI. Thirdly, we calculated the frequency of positions of each body part in grids for weighted MI. This step aims to make the position-correlated moving
--	--	--

		intensity of body part could be described by MI. Finally, the weighted MIs are plotted by heat map, which is easy to observe the movement area of each body part and depict the moving intensity in specific positions. Group effect When we compared the behavioral differences between KO and WT mice, we mainly evaluated them at the group level. We used two-way ANOVA to characterize the difference and draw conclusions. To demonstrate the behavior differences among individuals, the corresponding data in Fig. 6g are presented in Supplementary Table 3. We performed 100 pair-wise comparisons between 10 KO individuals and 10 WT individuals. We found that for hunching behavior, the probability in KO individuals is 87% higher than that in WT individuals; for the three subtypes of self-grooming behavior, the probability in KO individuals is higher than in WT individuals by 93%, 94%, and 96%, respectively. Combinatorial behaviors that present different clusters within the same group Profiling the behavioral patterns of transgenic animal disease models has critical significance. Besides comparing the behavioral difference with non-transgenic animals at the group level, there may also be subtypes with behavioral differences within the mutant group. However, for Shank3B KO mice, due to the limitations of previous behavior quantification methods, many studies have quantified behavior by human observation or velocity and position-based analysis. Among these studies, the most reported behavior maker is self-grooming^{20,21}, and a few studies mentioned differences in
--	--	--

		rearing behavior²². However, the subtypes of Shank3 and combinatorial behaviors are not reported. We have shown that our framework has the potential to discover new behavioral biomarkers. However, to further answer this question, we need to obtain a more detailed and comprehensive analysis as our next step. At this stage, we are focusing on demonstrating that our framework has the capacity for high-throughput analysis of behavioral data and investigation of behavioral differences. For example, in our newly added data (Supplementary Fig. 12 and Supplementary Table 2), we compared the behavioral patterns of the KO and WT groups under five different conditions. We found that for KO mice, changing the experimental apparatus, lighting conditions, ages, and sexes did not significantly affect the behavioral patterns; When the experimental conditions were the same, only the female groups of KO and WT had no significant difference. These findings are consistent with previous reports that Shank3B KO male mice display more severe impairments than females in motor coordination. - In Manuscript page 10-11, lines 340-356 (Discussion) Moreover, we further investigated the differences in the behavior patterns of Shank3B KO and WT mice at the group level. In addition to the data that had already been analyzed (collected under the condition: male mice, 5–6 weeks, white light, and circular open-field), we extended the group behavioral pattern analysis to include data collected under different conditions (i.e., different experimental apparatus, lighting, age, and gender; Supplementary Table 2). We calculated the cross-correlation coefficient matrix (CCCM) of all samples based on the movement fractions and used principal
--	--	--

		component analysis to extract the main variance factors of the CCCM (Supplementary Fig. 12 a, b). We found that when only a single condition was changed for male mice, there was no significant difference in population behavior patterns in mice with the same genotype (Supplementary Fig. 12 c). We also found that although some female KO mice had a weak tendency for autistic-like behavior, there was no significant difference between 5–6 week male and female KO mice at the group level when tested under the white-light circular open field condition (Supplementary Fig. 12 c, d). Finally, we compared the behavior patterns when all conditions were the same except for the genotypes. The results showed that only the female group showed no significant difference between KO and WT genotypes, while significant differences in behavioral patterns were found between KO and WT male mice under all other conditions. These findings are consistent with previous reports that Shank3B KO male mice display more severe impairments than females do in motor coordination.
R#3 (5)	I believe that one of the strength of this work is the decomposition of behaviour, which can then be tracked in time and used to predict modules of behaviour. I think that should this be applied extensively to behavioural studies will provide more convincing information about the validity. At the current state a longer monitoring in time, across days of individual animals would have provided, perhaps, a strongest validation of this framework.	Thank you for your suggestion. Continuously tracking the animal's poses and automatically decomposing and categorizing spontaneous behavior is particularly important for experiments that require long-term behavior observation. For example, when the timing of the expected behavior is hard to predict in advance, but the time window of the specific behavior is needed to locate it within the long-term recording to further analyze the correlation between the behavior and other measurements or for evaluating how chronic interventions such as drugs affect the animal's behavioral states over time. Therefore, according to your suggestion, we have added information on the continuous 24-hour recording experiment to provide further

		validation. The result shows that the strengths of our framework include:  1) Long-term accurate behavior tracking; 2) Efficient two-stage behavioral decomposition (especially for pose decomposition) for high-throughput behavior monitoring; 3) Cross-scale behavior quantification. - In Manuscript page 10, lines 322-327 (Discussion) For example, to study animal circadian rhythms, previous researchers have used electrophysiological and behavioral recording approaches to characterize different brain states. We used our framework to perform a continuous 24-hour behavioral recording, and the preliminary analysis proved that our framework could provide more comprehensive behavioral parameters and detailed quantification of behavior states (Supplementary Fig. 13). - In Supplementary Methods page 16, lines 337-366 (Supplementary Fig. 13 and figure legend) Supplementary Fig. 13 Continuous long-term monitoring and analysis of mouse behavior. a The timeline of the behavioral recording period over 24 hours. b The normalized velocity of the mouse across 24 hours aligned to the timeline. c The decomposed behavioral modules shown with color-coded labels. d Three magnified representative behavioral modules and selected, single corresponding frames. Left, running on the litter; Middle, eating; Right, prolonged immobility resembling resting. e, f State transitions of the movement modules in night and day phases. g Differences in the state transitions between night and day. The color of the dots in e, f, and g correspond to the behavioral modules shown in c. The size of
--	--	--

		the dots represents the rank of the module probabilities over 24 hours. The color of the connections in e and f represents the direction from the previous state to the current state, and its color is the same as that of the previous state. The width of the connections in e and f represents the normalized two-state transition probability. The color and width of the connections in g represent the normalized difference between e and f. - In Supplementary Methods page 19, lines 437-4445 (Supplementary Methods Animals, behavioral experiments and behavioral data collection) In the second behavioral experiment, we capture the mouse behavior for 24 hours (related to Supplementary Fig. 13). We still use circular open field but covered by wood chip as padding and offered water and regular food (the chow). The male mouse used in this experiment has C57BL/6J genetic background and is 13 weeks old. To change the light conditions and keep the circadian rhythms of mouse, we use infrared light as the background light and set the cameras to infrared model. This experiment was start at 20:20 p.m. We first closed the light until 7:00 a.m. next day, then we open the white light until 19:00 p.m. At last, we closed the light until 20:20 p.m. and finished the behavioral capturing across 24 hours. All the detailed information of mice and experimental conditions are in Supplementary Table 2
R#3 (6)	One more last thing, although it is outside of the scope of this study, are the authors planning to extend this framework to mouse social interaction? As I said, this won't change what they have nicely	In fact, we are also very interested in both interspecies or intraspecies social behaviors, such as mating, social hierarchy, predation, and defense behaviors. Currently, extending our framework to social behavior analysis is limited by the problem of tracking multiple visually indistinguishable (marker less) animals without their

	achieved in this study, it is just a curiosity and an interest that involve everyone in the community.	identities being swapped. Ideally, if the poses of multiple animals can be accurately estimated in each frame, then based on the temporal context (smooth trajectories), these animals can be correctly tracked²³. However, once the animal's bodies are touching or even occluded, the inaccurate poses estimation leads to off-tracking and identity-swap. Although recent works include the DLC for multi-animal pose estimation²⁴, SLEAP²⁵ and AlphaTracker²⁶, this issue haven't been solved perfectly. Another approach for social interacting animal tracking is from the population aspect, which only requires the features unrelated to animals' identities, such as using the position differences between animals' body parts for recognition or clustering. However, this approach is limited to specific behaviors and does not apply to unequal status behaviors between social subjects. Our 3D multi-view motion capture system is promising method to solve this problem since it can effectively reduce body occlusion probability. We have tested the tracking of animals with different appearances, as well as animals with similar appearances without body touching. The results demonstrated that our system tracks them perfectly (Supplementary Video 6, 7). However, it performs poorly for animals with similar appearances and body touching. This is because when estimating multiple body parts of multiple animals in a single frame, the combination of the poses of these animals is more complex and diverse, and the identity-swap in different views may happen at different times. To this end, we are considering using computer vision technology (e.g., point cloud reconstruction) to fuse images from multi-views, then segment each animal's body, and estimate the body parts based on the reconstructed 3D animal. If these problems can be solved well, we will expand our framework,
--	---	---

		including adding modules to analyze social behaviors, and using our framework to generate a large number of well-annotated behavior sample databases. We believe that the development of these toolboxes will be definitely beneficial to the community. We added the discussion of the limitations of the framework and its future direction: - In Manuscript page 11-12, lines 376-394 (Discussion) Lastly, we would like to discuss the limitations of our framework. When extending our framework to social behavior analysis, such as the analysis of mating, social hierarchy, predation, and defense behaviors, it is challenging to track multiple, visually indistinguishable (markerless) animals without identity-swapping errors (Supplementary Video 6, 7). Alternative methods mainly focus on tracking and identifying social behaviors at the population level, which only requires the identification of features unrelated to the animals' identities such as the positional differences between animals' body parts. However, this approach is limited to specific behaviors and does not apply to interaction behaviors between social subjects of unequal status. Recent cutting-edge toolboxes such as DLC for multi-animal pose estimation, SLEAP, and AlphaTracker have addressed the multi-animal tracking problem, but once animals with similar appearances are touching or even body-occluded, the inaccurate pose estimation of these toolboxes leads to off-tracking and identity-swapping errors. This is because when estimating multiple body parts of several animals in a single frame, the combination of the poses of these animals is more complex and diverse, and identity-swapping in different views may happen at different times. Our 3D multi-view motion capture system promises to solve this problem by effectively
--	--	--

		reducing body-occlusion probability. As a next step, we are considering using computer vision technology (e.g., point cloud reconstruction) to fuse images from multiple views, then segment each animal's body, and estimate the body parts based on the reconstructed 3D animal. Solving these problems will extend the applicability of our framework to the benefit of the animal behavioral research community.
--	--	---

References

1. Singh, S., Bermudez-Contreras, E., Nazari, M., Sutherland, R. J. & Mohajerani, M. H. Low-cost solution for rodent home-cage behaviour monitoring. *PLoS ONE* **14**, 1–18 (2019).
2. Bains, R. S. *et al.* Assessing mouse behaviour throughout the light/dark cycle using automated in-cage analysis tools. *Journal of Neuroscience Methods* **300**, 37–47 (2018).
3. Gris, K. V., Coutu, J. P. & Gris, D. Supervised and unsupervised learning technology in the study of rodent behavior. *Frontiers in Behavioral Neuroscience* **11**, 1–6 (2017).
4. Datta, S. R., Anderson, D. J., Branson, K., Perona, P. & Leifer, A. Computational Neuroethology: A Call to Action. *Neuron* **104**, 11–24 (2019).
5. Markowitz, J. E. *et al.* The Striatum Organizes 3D Behavior via Moment-to-Moment Action Selection. *Cell* **174**, 44-58.e17 (2018).
6. von Ziegler, L., Sturman, O. & Bohacek, J. Big behavior: challenges and opportunities in a new era of deep behavior profiling. *Neuropsychopharmacology* 1–12 (2020) doi:10.1038/s41386-020-0751-7.
7. Hinde, R. A. & Bateson, P. P. G. *Growing Points Ethology*. (CUP Archive, 1976).
8. Keogh, E. & Ratanamahatana, C. A. Exact indexing of dynamic time warping. *Knowledge and Information Systems* **7**, 358–386 (2005).
9. Shimodaira, H., Nakai, M., Noma, K. & Sagayama, S. Dynamic Time-Alignment Kernel in Support Vector. *Nips* (2001).

10. Wang, Y., Huang, H., Rudin, C. & Shaposhnik, Y. Understanding How Dimension Reduction Tools Work: An Empirical Approach to Deciphering t-SNE, UMAP, TriMAP, and PaCMAP for Data Visualization. 1–63 (2020).
11. Kobak, D. & Berens, P. The art of using t-SNE for single-cell transcriptomics. *Nature Communications* **10**, (2019).
12. Valletta, J. J., Torney, C., Kings, M., Thornton, A. & Madden, J. Applications of machine learning in animal behaviour studies. *Animal Behaviour* **124**, 203–220 (2017).
13. Rousseeuw, P. J. Silhouettes: A graphical aid to the interpretation and validation of cluster analysis. *Journal of Computational and Applied Mathematics* **20**, 53–65 (1987).
14. Scrucca, L., Fop, M., Murphy, T. B. & Raftery, A. E. *mclust 5: Clustering, Classification and Density Estimation Using Gaussian Finite Mixture Models*. <http://cran-logs.rstudio.com>.
15. Wang, X. *et al.* Synaptic dysfunction and abnormal behaviors in mice lacking major isoforms of Shank3. *Human Molecular Genetics* **20**, 3093–3108 (2011).
16. Merel, J., Botvinick, M. & Wayne, G. Hierarchical motor control in mammals and machines. *Nature Communications* **10**, 1–12 (2019).
17. Churchland, M. M. & Shenoy, K. V. Temporal complexity and heterogeneity of single-neuron activity in premotor and motor cortex. *Journal of Neurophysiology* **97**, 4235–4257 (2007).
18. Scott, S. H. Inconvenient Truths about neural processing in primary motor cortex. *Journal of Physiology* **586**, 1217–1224 (2008).
19. Gao, P. & Ganguli, S. On simplicity and complexity in the brave new world of large-scale neuroscience. *Current Opinion in Neurobiology* **32**, 148–155 (2015).
20. Peça, J. *et al.* Shank3 mutant mice display autistic-like behaviours and striatal dysfunction. *Nature* **472**, 437–442 (2011).
21. Sukoff Rizzo, S. J. & Crawley, J. N. Behavioral Phenotyping Assays for Genetic Mouse Models of Neurodevelopmental, Neurodegenerative, and Psychiatric Disorders. *Annual Review of Animal Biosciences* **5**, 371–389 (2017).
22. Peixoto, R. T. *et al.* Abnormal Striatal Development Underlies the Early Onset of Behavioral Deficits in Shank3B^{-/-} Mice. *Cell Reports* **29**, 2016-2027.e4 (2019).
23. Robie, A. A., Seagraves, K. M., Egnor, S. E. R. & Branson, K. Machine vision methods for analyzing social interactions. *Journal of Experimental Biology* **220**, 25–34 (2017).
24. Mathis, A. *et al.* DeepLabCut: markerless pose estimation of user-defined body parts with deep learning. *Nature Neuroscience* **21**, 1281–1289 (2018).
25. Pereira, T. D. *et al.* SLEAP: Multi-Animal Pose Tracking. *bioRxiv* 2020.08.31.276246 (2020) doi:10.1101/2020.08.31.276246.

26. Chen, Z. *et al.* AlphaTracker: A Multi-Animal Tracking and Behavioral Analysis Tool. *bioRxiv* 2020.12.04.405159 (2020)
doi:10.1101/2020.12.04.405159.

Reviewers' Comments:

Reviewer #1:

Remarks to the Author:

The reviewer made comments in the last round and criticize the novelty of the proposed framework in this paper.

After having read the revised version, the reviewer has NOT been convinced by the novel contribution made in the proposed 3d motion capture system.

The reviewer believes that the proposed motion capture system is an incremental version of individual standard technologies, including camera calibration, pose estimation, trajectory estimation, etc.

The revision does not justify the number of mice used in the experiments. Therefore, the statistics generated in the current version are not convincing.

Again, the authors fail to persuade the reader of why the existing mouse motion capture systems should not be used in the community.

The created images/figures in the paper look too rush.

Reviewer #3:

Remarks to the Author:

The authors answered all my questions and I appreciated the new data on social interaction!

Responses to the Review Comments (NCOMMS-20-43913A)

We wish to thank the reviewer for their thoughtful comments of our revised submission. These inputs prompted us to improve our manuscript. This document provides a point-by-point response to the comments raised by the review. We believe the revised paper is better positioned, more focused, and makes a stronger contribution to the literature. We sincerely hope that you will find that this revised manuscript has improved substantially and is heading in the right direction. All other changes made to the Manuscript are highlighted by using the track changes.

Reviewer #1 (R#1)		
COMMENT No.	REVIEW COMMENTS	AUTHORS' RESPONSES
R#1 (1)	After having read the revised version, the reviewer has NOT been convinced by the novel contribution made in the proposed 3d motion capture system. The reviewer believes that the proposed motion capture system is an incremental version of individual standard technologies, including camera calibration, pose estimation, trajectory estimation, etc.	Thank the reviewer for the comment. Our main contribution in this work is providing a low-cost and efficient 3D motion capture system solution. The closed-loop auto-calibration module has not been used in the literature. This approach can acquire the checkerboard images fast and significantly improve the success rate of calibration. We provide the detailed validations of body occlusion in multi-view, and multi-body parts 3D tracking and cameras setting. Besides, obtaining high-quality 3D animal behavior data is only the first step. We also propose and implement a novel behavior natural structure-inspired decomposition framework and use this technology to collect and characterize the behavior of the animal disease model. Our data proved that our framework could detect behavioral biomarkers that have been identified previously and discover potential new behavioral biomarkers. We will continue to develop related techniques to promote our lab and the potential involved users to collect and annotate the data to build the animal behavior dataset. The motivation for this work is from our previous behavioral paradigms and analytical tools could not answer many questions well. For example, our lab and other research^{1,2} groups interested in visually evoked innate defensive behavior. We used the looming stimulus to simulate the threaten approaching, and the mouse will show the stereotyped defensive response. Due to the limitations of the behavioral measurement approaches, we can only use the freezing or flighting response to represent animals fear

		emotion states (these two behaviors are easy to quantify). However, in our behavioral videos, we observed rich behavioral patterns after the stimulus releasing (for example, the animal usually shows rearing or orienting actions before freezing or flighting). Some reviews also proposed a similar conceptual timeline of events during escape behavior³. Current available behavioral study tools struggle to identify these behavioral patterns, both the commercial solutions or open-source toolboxes. To this end, our primary demand is to extract biologically significant behavior sequence modules from continuous behavioral videos. So, we first proposed the two-stage dynamic temporal decomposition algorithm to address the high-dimensional and complex rodent behavior. For the pose trajectories acquiring, the original solution is DLC-based 2D pose estimation. Since the looming stimulus should be delivered in the upper filed, it's inconvenient to capture the behavior video in the top-view. The side-view 2D DLC data caused heterogeneous behavioral decomposition since heavy body-occlusions and view-angle bias (highly similar actions from different view-angle were captured and identified as different actions). Thus, we had to find 3D solutions. The DLC and existing 3D reconstruction methods are well-developed (except tracking multi-animal with similar appearances), but they are individual technical modules. As we mentioned in the previous response letter, we made many efforts to integrate and optimize the 3D motion capture system to enable highly efficient and high-quality 3D behavior data collection. Besides the auto-calibration and synchronous multi-video acquiring, the validation and pre-processing of the 3D skeleton, including performance validation, body part definition, body alignment, and body scaling, is also essential for 3D behavior analysis⁴. We believe that our practice will be helpful for many researchers in this field.
--	--	--

R#1 (2)	The revision does not justify the number of mice used in the experiments. Therefore, the statistics generated in the current version are not convincing.	Thank the reviewer for the comment. The sample sizes of behavior tests were selected by referring to the previous behavior studies of Shank3B mutant mice ⁵⁻⁸:  - Peça, J. et al., Nature, 2011. Figure 1, n = 6-9 mice per genotype; - Mei Y. et al., Nature, 2016. Figure 1, n_{WT} = 6, n_{KO} = 7 mice; - Wang X. et al., Human Molecular Genetics, 2011. Figure 2-4, n = 10 mice/genotype/sex; - Fourie, C. et al., Frontiers in Cellular Neuroscience, 2018. Figure 1, n_{WT} = 8, n_{KO} = 10 mice; Besides, the sample size selection is also verified by power analysis ⁹.
R#1 (3)	Again, the authors fail to persuade the reader of why the existing mouse motion capture systems should not be used in the community.	Thank the reviewer for the comment. 3D behavior monitoring is crucial for the mammal's spontaneous behavior study. Datta et al. held that it is easy to understand that the animals naturalistically explore the world through 3D movements of their bodies and so express pose dynamics in 3D ¹⁰. Animals show rich kinematical information through their body (especially the nose, head, limbs and trunk) in many movements (e.g., rearing, stretching, climbing). Capturing 3D motion information makes it reliable when identifying these behaviors ¹¹. In addition, the 3D skeletal data allows a detailed description of the behavioral types. In our manuscript, we compared the newly identified hunching with rearing movement by providing the 3D motion "portraits". Thus, we proved the hunching is a unique behavior and different from rearing.

		Currently, we are working on conducting a comprehensive comparison between our framework and traditional behavior analysis approach for evaluating the anxiety-like mouse models' behavior (unpublished). The preliminary results showed that compared with the single-camera solution, our approach can significantly identify the behavioral differences of the anxiety-like mice, whereas the traditional method could not detect the significance. Finally, we would like to clarify that to replace existing behavioral capture systems, more development is needed for the 3D behavior collection system (e.g., the novel and validated 3D behavior paradigms). We added the discussion of this point. - In Manuscript page 10, lines 319-323 (Discussion): Currently, we are working on conducting a comprehensive comparison between our framework and traditional behavior analysis approach for evaluating the anxiety-like mouse models' behavior. The preliminary results showed that compared with the single-camera solution, our approach can significantly identify the behavioral differences of the anxiety-like mice, whereas the traditional method could not detect the significance.
R#1 (4)	The created images/figures in the paper look too rush.	In this revision, we have carefully checked our figures according to the editor's instructions. We have improved the figure legends, axes labels, abbreviations, symbols and colors. We hope these improvements are heading in the right direction.

References

1. Wei, P. *et al.* Processing of visually evoked innate fear by a non-canonical thalamic pathway. *Nature Communications* **6**, 6756 (2015).
2. Shang, C. *et al.* Divergent midbrain circuits orchestrate escape and freezing responses to looming stimuli in mice. *Nature Communications* **9**, 1232 (2018).
3. Evans, D. A., Stempel, A. V., Vale, R. & Branco, T. Cognitive Control of Escape Behaviour. *Trends in Cognitive Sciences* **23**, 334–348 (2019).
4. von Ziegler, L., Sturman, O. & Bohacek, J. Big behavior: challenges and opportunities in a new era of deep behavior profiling. *Neuropsychopharmacology* **46**, 33–44 (2021).
5. Peça, J. *et al.* Shank3 mutant mice display autistic-like behaviours and striatal dysfunction. *Nature* **472**, 437–442 (2011).
6. Mei, Y. *et al.* Adult restoration of Shank3 expression rescues selective autistic-like phenotypes. *Nature* **530**, 481–484 (2016).
7. Wang, X. *et al.* Synaptic dysfunction and abnormal behaviors in mice lacking major isoforms of Shank3. *Human Molecular Genetics* **20**, 3093–3108 (2011).
8. Fourie, C. *et al.* Dietary Zinc Supplementation Prevents Autism Related Behaviors and Striatal Synaptic Dysfunction in Shank3 Exon 13–16 Mutant Mice. *Frontiers in Cellular Neuroscience* **12**, 1–14 (2018).
9. Rosenthal, R. Parametric measures of effect size. in *The handbook of research synthesis* 231–244 (1994).
10. Datta, S. R. Q&A: Understanding the composition of behavior. *BMC Biology* **17**, 1–7 (2019).
11. Marshall, J. D. *et al.* Continuous Whole-Body 3D Kinematic Recordings across the Rodent Behavioral Repertoire. *Neuron* **109**, 420-437.e8 (2021).